# DOT: a flexible multi-objective optimization framework for transferring features across single-cell and spatial omics

Arezou Rahimi[1,2], Luis A. Vale-Silva [2], Maria Fälth Savitski[2], Jovan Tanevski [1,3,4] ✉ & Julio Saez-Rodriguez [1,4] ✉

Single-cell transcriptomics and spatially-resolved imaging/sequencing technologies have revolutionized biomedical research. However, they suffer from lack of spatial information and a trade-off of resolution and gene coverage, respectively. We propose DOT, a multi-objective optimization framework for transferring cellular features across these data modalities, thus integrating their complementary information. DOT uses genes beyond those common to the data modalities, exploits the local spatial context, transfers spatial features beyond cell-type information, and infers absolute/relative abundance of cell populations at tissue locations. Thus, DOT bridges single-cell transcriptomics data with both high- and low-resolution spatially-resolved data. Moreover, DOT combines practical aspects related to cell composition, heterogeneity, technical effects, and integration of prior knowledge. Our fast implementation based on the Frank-Wolfe algorithm achieves state-of-the-art or improved performance in localizing cell features in high- and low-resolution spatial data and estimating the expression of unmeasured genes in low-coverage spatial data.

The organization of cells within human tissues, their molecular programs and their response to perturbations are central to better understanding physiology, disease progression and the eventual identification of targets for therapeutic intervention[1,2]. Single-cell RNA sequencing can profile a large part of the transcriptome of many individual (single) cells. This has made this technology (hereafter scRNA-seq) an essential tool for revealing distinct cell features (such as cell lineage and cell states) in complex tissues and has profoundly impacted our understanding of biological processes and the underlying mechanisms that control cellular functions[3–5]. However, scRNA-seq requires dissociation of the cells[6], losing the information about their spatial context and physical relationships, which is critical to understand the functioning of tissues.

To overcome this limitation, there have been recent advancements in spatially resolved transcriptomics (SRT) methods[7–9]. SRT methods measure gene expression in locations coupled with their two- or three-dimensional position. SRT methods vary in two axes: spatial resolution and gene coverage. On one hand, technologies such as multiplexed error-robust fluorescence in situ hybridization (MERFISH) and in situ sequencing (ISS), achieve cellular or even subcellular resolution[10], but are limited to measuring up to a couple of hundred pre-selected genes. On the other hand, spatially resolved RNA sequencing, such as Spatial Transcriptomics[11], commercially available as 10X's Visium, and Slide-seq[12], enable high-coverage gene profiling by capturing mRNAs in-situ but come at the cost of measuring these averaged within spots that include multiple cells. Thus, there is a trade-off between resolution and richness (gene coverage) of SRT data.

A natural strategy to provide a complete picture is to enrich SRT by transferring features, such as cell types/states or the expression of unmeasured genes, from scRNA-seq to spatial locations in SRT.

[1]Institute for Computational Biomedicine, Heidelberg University & Heidelberg University Hospital, Heidelberg, Germany. [2]Cellzome GmbH, GlaxoSmithKline, Heidelberg, Germany. [3]Department of Knowledge Technologies, Jožef Stefan Institute, Ljubljana, Slovenia. [4]These authors contributed equally: Jovan Tanevski, Julio Saez-Rodriguez. ✉e-mail: jovan.tanevski@uni-heidelberg.de; pub.saez@uni-heidelberg.de

However, integrating scRNA-seq and SRT is often challenging for many reasons, such as the limited number of genes shared across these modalities, differences in measurement sensitivities across technologies, and high computational cost for large-scale datasets. Computational methods in recent years have attempted to address these challenges from different perspectives[13–16]. The vast majority of these methods are dedicated to cell-type deconvolution in low-resolution spatial data. Along this vein, SPOTlight[17] uses non-negative matrix factorization to factorize the scRNA-seq count matrix, thereby modeling the gene expressions in spots and determining their cell-type compositions. Similarly, CARD[18] employs non-negative matrix factorization and the spatial correlation structure in cell-type composition across tissue locations. Moreover, statistical methods such as RCTD[19] and cell2location[20] are built upon the hypothesis of similarity of cell-type specific distribution of gene expression in single-cell and spatial transcriptomics. On the other hand, methods such Tangram[21], TACCO[22], and CytoSpace[23], have been proposed to integrate scRNA-seq and high-resolution SRT.

Here, we present an approach, DOT, that addresses the limitations and challenges of both groups of methods. DOT is a versatile and scalable optimization framework for the integration of scRNA-seq and SRT for localizing cell features by solving a multi-criteria mathematical program. In addition to being applicable to various types of spatial omics, a distinctive feature of DOT is that it exploits the spatial context and the genes that are present in scRNA-seq or SRT but missing in the other. This is in contrast to approaches that do not use the spatial localization information in the spatial omics and rely on the genes that are mutually captured by both scRNA-seq and SRT without using the remaining genes exclusively captured in each modality.

One major challenge that we address with DOT is taking into consideration the spatial context of the data explicitly. On the one hand, neglecting the spatial context is equivalent to assuming random placement of spots in the space, which is at odds with the established structure-function relationship of tissues[9]. On the other hand, assuming a uniform dependence between cell-type composition and spatial location across different regions of the tissue might not hold for complex tissues. In contrast, DOT leverages the spatial context in a local manner without assuming a global correlation. Our local view of spatial features allows us to utilize them only when it is beneficial to do so, based on a threshold on the similarity of gene expression of adjacent spots. Another feature of DOT is that it exploits the genes that are not mutually measured in scRNA-seq and SRT. Considering only a subset of genes limits the applicability of these methods to cases where the two data sets share several informative genes, which might not be the case when different technologies are used for profiling, or when few genes are measured in the spatial data (e.g., in MERFISH). In DOT, we use the genes exclusively measured in scRNA-seq to capture the heterogeneity of cell populations by sub-clustering them into refined clusters, and use the distinct genes in SRT to inform the locally relevant spatial structures.

Another distinctive feature of DOT is that it is applicable to both high- and low-resolution SRT, as our model is capable of inferring membership probabilities for the former and the absolute abundance of cell populations and size of spots in the latter. Additionally, DOT works with both discrete counts and continuous expressions. This distinguishes our model from optimal transport-based models (such as TACCO) and deep learning methods (such as Tangram), which do not offer absolute abundances in low-resolution, and statistical methods (such as cell2location and RCTD), which rely on discrete mRNA counts.

Our optimization model considers several practical aspects in a unified framework, including (i) local spatial relations between different cell features, (ii) differences in measurement sensitivity of different technologies, (iii) heterogeneity of cell populations, (iv) compositional sparsity and size of spatial locations at different spatial resolutions, and (v) incorporation of prior knowledge about the expected abundance of cell features in situ. We present a fast solver-free solution based on the Frank-Wolfe algorithm, thereby ensuring scalability and efficiency for large-scale datasets. DOT has a broader application beyond cell-type decomposition, including transferring continuous features such as the expression of genes that are missing in SRT but present in scRNA-seq data. DOT is freely available to facilitate its application and further development. The data-driven self-adapting nature of DOT further facilitates its deployment with minimal user involvement.

## Results

### DOT maps cell features to space by multi-objective optimization

Given a reference scRNA-seq data (R for short), which is a collection of single cells each annotated with a categorical or continuous feature (such as cell type or cell state), and a target SRT data (S for short), which consists of a set $\mathbb{I}$ of spots, associated with a location containing one or more cells, we wish to determine the abundances (in the case of multiple cells per spot) or single value (in the case of a single-cell per spot) of the unobserved feature(s) in spots of S (see Fig. 1). In what follows, we assume that the unobserved features are categorical values in a set $\mathbb{C}$ and note that the continuous case extends naturally. Consequently, we assume that the cells in R are categorized into $|\mathbb{C}|$ cell populations.

Our mathematical model relies on determining a "many-to-many" mapping (transfer) $\mathbf{Y}$ of cell populations in R to spots in S, with $Y_{c,i}$ denoting the abundance of cell population $c \in \mathbb{C}$ in spot $i \in \mathbb{I}$. When S is high resolution (i.e., each spot is a cell), $Y_{c,i}$ determines the probability that spot $i \in \mathbb{I}$ belongs to cell population $c \in \mathbb{C}$, whereas $Y_{c,i}$ determines the number of cells in spot $i$ that belongs to cell population $c \in \mathbb{C}$ when S is low-resolution (i.e., spots are composed of multiple cells).

Let $X^{R}_{c,g}$ and $X^{S}_{i,g}$ denote the expression profiles of cell population $c \in \mathbb{C}$ and spot $i \in \mathbb{I}$, respectively, for genes $g \in \mathbb{G}$. We assume that $X^{R}_{c,g}$ is the mean expression of gene $g$ across the cells that belong to cell population $c \in \mathbb{C}$ of R (see Methods for extension to heterogeneous cell populations). Moreover, $X^{S}_{i,g}$ is the aggregation of expression profiles of potentially several cells when S is low-resolution. A high-quality transfer should naturally match the expression of the common genes across R and S. We ensure this by considering the following expression-focused criteria:

(i) Matching expression profile of spots (Fig. 1b). Expression profile of each spot $i \in \mathbb{I}$ in S (i.e., $\mathbf{X}^{S}_{i,:}$) should match the expression profile transferred to that spot from R via $\mathbf{Y}$ (i.e, $\sum_{c \in \mathbb{C}} Y_{c,i} \mathbf{X}^{R}_{c,:}$). We penalize the dissimilarity of these vectors via:

$$d_{i}(\mathbf{Y}) := d_{\cos}\left(\mathbf{X}^{S}_{i,:}, \sum_{c \in \mathbb{C}} Y_{c,i} \mathbf{X}^{R}_{c,:}\right). \tag{1}$$

(ii) Matching expression profile of cell populations (Fig. 1c). Expression profile of each cell population $c \in \mathbb{C}$ in R should match the expression profile of spots assigned to this cell population via $\mathbf{Y}$:

$$d_{c}(\mathbf{Y}) := d_{\cos}\left(\mathbf{X}^{R}_{c,:}, \sum_{i \in \mathbb{I}} Y_{c,i} \mathbf{X}^{S}_{i,:}\right). \tag{2}$$

(iii) Matching spatial gene expression maps (Fig. 1d). Spatial expression map of each gene $g \in \mathbb{G}$ in S should be similar to the expression map of that gene as transferred from R via $\mathbf{Y}$:

$$d_{g}(\mathbf{Y}) := d_{\cos}\left(\mathbf{X}^{S}_{:,g}, \sum_{c \in \mathbb{C}} \mathbf{Y}_{c,:} X^{R}_{c,g}\right). \tag{3}$$

In the above formulations, $d_{\cos}$ is a scale-invariant metric based on cosine-similarity that measures the difference between two vectors regardless of their scales (Methods).

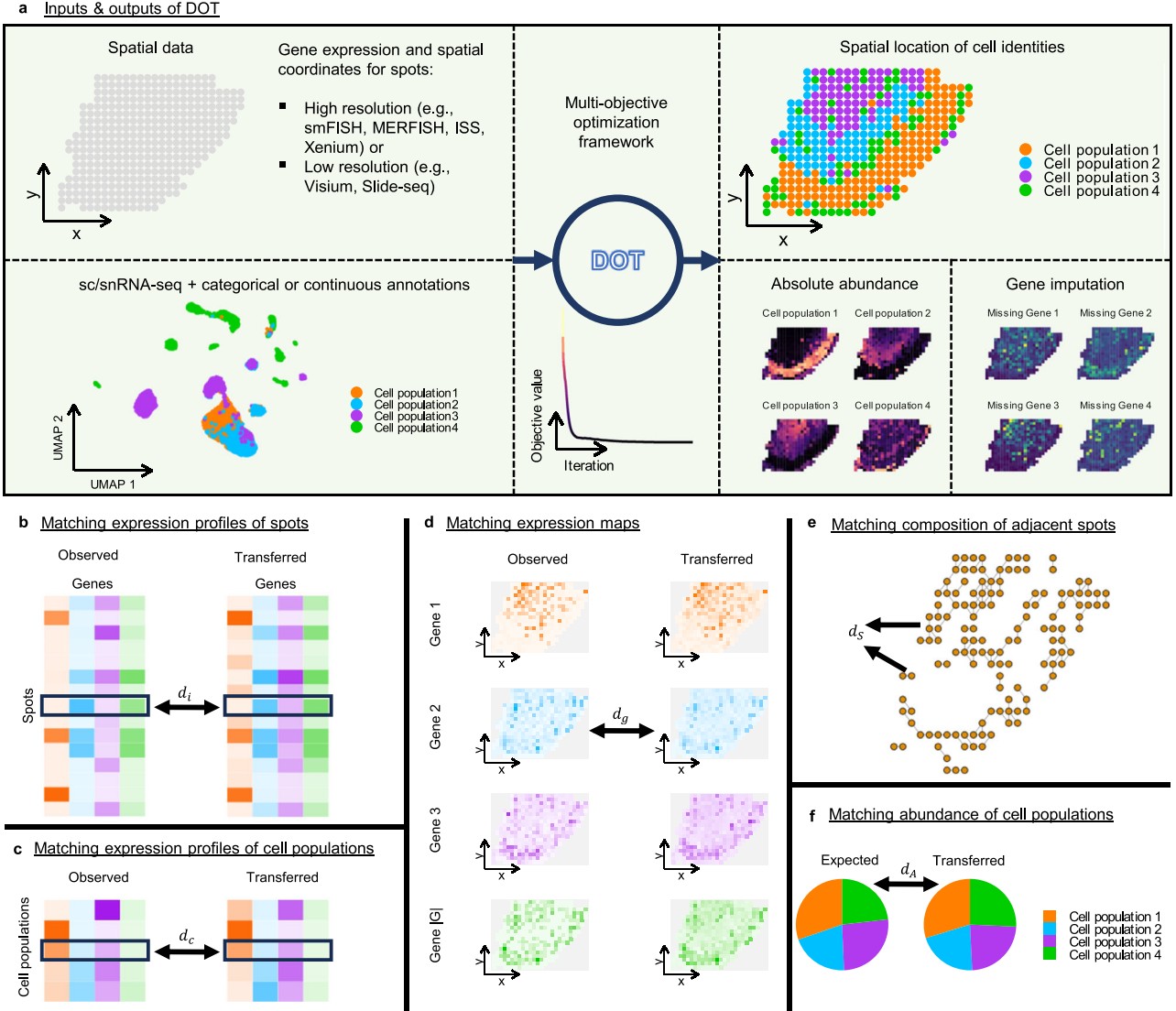

**Fig. 1 | Overview of inputs and outputs of DOT and its optimization framework.**
**a** From left to right: DOT takes two inputs: (i) spatially resolved transcriptomics data, which contains spatial measurements of genes at either high or low-resolution spots and their spatial coordinates, and (ii) reference singe-cell RNA-seq data, which contains single cells with categorical (e.g., cell type) or continuous (e.g., expression of genes that are missing in the spatial data) annotations. DOT employs several alignment objectives to locate the cell populations and the annotations therein in the spatial data. The alignment objectives ensure a high-quality transfer from different perspectives: **b** the expression profile of each spot in the spatial data (left) must be similar to the expression profile transferred to that spot from the reference scRNA-seq data (right), **c** the expression profile of each cell population in the reference data (left) must be similar to the expression profile of that cell population inferred in the spatial data (right), **d** expression map of each gene in the spatial data (left) must be similar to expression map of that gene as transferred from the reference data (right), **e** spots that are both adjacent and have similar expression profiles are likely to have similar compositions, and (**f**) if prior knowledge about the expected relative abundance of cell populations is available, the transfer should retain the given abundances.

In addition to the expression-focused objectives, we introduce optional constraints related to the similarity of expressions in neighboring spatial locations of spots as well as introduce prior knowledge in the form of the expected abundance of cell populations using the following *compositional* criteria:

(iv)    Capturing spatial relations (Fig. 1e). Spots that occupy adjacent locations and have similar expression profiles are expected to be of similar compositions. Given $\mathbb{P}$, the set of adjacent pairs of spots with similar expression profiles, we encourage similar composition profiles for these spots by penalizing

$$d_S(\mathbf{Y}) := \sum_{(i,j)\in\mathbb{P}} w_{ij}d_{JS}(\mathbf{Y}_{:,i},\mathbf{Y}_{:,j}), \tag{4}$$

where $d_{JS}$ is the Jensen-Shannon divergence and $w_{ij}$ captures the similarity of expression profiles of spots $i$ and $j$ (Methods).

(v)    Matching expected abundances (Fig. 1f). If prior information about the expected abundance of cell categories in S is available (e.g., when R and S correspond to adjacent tissues or consecutive sections), then the abundance of cell categories transferred to S should be consistent with the given abundances. We measure dissimilarity between the vector of expected abundances (denoted **r**) and abundance of cell categories in S via

$$d_A(\mathbf{Y}) := d_{JS}(\mathbf{Ye},\mathbf{r}). \tag{5}$$

The expression-focused objectives naturally take precedence over the compositional objectives, especially when a large number of genes are common between R and S, but the compositional objectives are useful when the number of common genes is limited. Note that objective (v) provides additional control over the abundance of cell types in S, but can be ignored if prior information about the abundance of cell types is not available.

We treat these criteria as objectives in a multi-objective optimization problem, and to consider them simultaneously (i.e., produce a Pareto-optimal solution), we optimize **Y** against a linear combination of these objectives as formulated below, hereafter referred to as the DOT model:

$$\min \quad \sum_{i\in\mathbb{I}} d_i(\mathbf{Y}) + \lambda_C \sum_{c\in\mathbb{C}} d_c(\mathbf{Y}) + \lambda_G \sum_{g\in\mathbb{G}} d_g(\mathbf{Y}) + \lambda_S d_S(\mathbf{Y}) + \lambda_A d_A(\mathbf{Y}), \quad (6)$$

$$\text{w.r.t.} \quad \mathbf{Y} \in \mathbb{R}_+^{|\mathbb{C}|\times|\mathbb{I}|}, \quad (7)$$

$$\text{s.t.} \quad 1 \le \sum_{c\in\mathbb{C}} Y_{c,i} \le n_i \quad \forall i\in\mathbb{I}. \quad (8)$$

Here, $\lambda_C$, $\lambda_G$, $\lambda_S$, and $\lambda_A$ are the user-defined penalty weights, and $n_i$ is an upper bound on the expected size (number of cells) of spot $i\in\mathbb{I}$ (i.e., $n_i=1$ for high resolution SRT). For low-resolution SRT, we set $n_i=n$ for a pre-determined parameter $n$ and let the model determine the size of the spots (Methods).

Next, we present an evaluation of the model, comparing its performance to the related work and highlighting different aspects of DOT in different applications. Briefly, we evaluate the performance of DOT to transfer the cell type label of single-cell level spots in high-resolution SRT decompose spots to cell type abundances in low-resolution SRT, and estimate the expression of genes that are missing in SRT but are measured in the reference scRNA-seq. Details of the datasets and performance metrics used for these experiments are presented Methods and Supplementary Notes.

## DOT locates cell types in high-resolution spatial data
Our goal with our first set of experiments is to evaluate the performance of different models in determining the abundance of cell types at each spot. We used the high-resolution MERFISH spatial data of the primary motor cortex region (MOp) of the mouse brain[24], which contains the spatial information and cell type of 280,186 cells across 64 samples (Supplementary Notes). Since the cell type represented in the spot is known in our high-resolution spatial data, we can use this information as ground truth when evaluating the performance of the different models. Details about the benchmark instances can be found in Methods.

We compared the performance of DOT against a total of seven methods, including four models from the literature for transferring cell types from single-cell to high-resolution SRT: NovoSpaRc[25], Tangram[21], TACCO[22], and SingleR[26]. As baseline, we used two general-purpose methods from the literature. Namely, we used the anchoring method from Seurat[27], which builds a mutual nearest neighbors graph between the cells in the reference scRNA-seq data and the spots in the target SRT data, thereby inferring a membership probability for the cell type composition of the spots. In addition, we used the single-cell integration method from Harmony[28]. More specifically, we mapped both scRNA-seq and SRT into a shared PC space, and assigned the 10 nearest cells to each spot based on their Euclidean distance in the latent space. We then inferred the cell type composition of each spot based on the cells assigned to that spot. Finally, given the multiclass classification nature of cell type prediction in high-resolution SRT, we also used Random Forests (RF)[29] as a multiclass classifier baseline. For completeness, we provide the performance of other methods

(including cell2location[20], RCTD[19], and CytoSpace[23]) in Supplementary Fig. 1.

DOT clearly outperforms the six specialized methods and the baseline classification method in assigning correct cell types to the spots (first row in Fig. 2). DOT produces well-calibrated probabilities (second row in Fig. 2) and better captures the relationships between cell types in space (third row in Fig. 2). This is a result of its capacity to incorporate the spatial information through $d_S$. We also observe that even with very few genes in common between SRT and the reference scRNA-seq data (e.g., $|\mathbb{G}|\le 75$), DOT is able to reliably determine the cell type of spots in the space with high accuracy. In contrast, most benchmark methods fail to produce results due to a lack of shared information or produce results with lower accuracy. Finally, we observed that DOT performs robustly under fluctuations in gene expression. Each column in Fig. 2 corresponds to a different level of additive uniform noise to the expression of the considered genes in the range of 0 to 50%.

## DOT determines cell type abundances in low-resolution spatial data
Since there is no ground truth for real low-resolution spatial data such as Visium and Slide-seq, we produce ground truth low-resolution spatial data in an objective manner by reproducing measurements of low-resolution data by pooling adjacent cells in the high-resolution spatial data of primary motor cortex of the mouse brain (MOp), primary somatosensory cortex of the mouse brain (SSp), and the developing human heart. Figure 3a illustrates a sample low-resolution SRT obtained from the high-resolution MERFISH data of a MOp tissue.

In Fig. 3b we show the comparison of the performance of DOT against ten methods, including cell2location (C2L)[20], CARD[18], CytoSpace[23], Harmony, NovoSpaRc, RCTD[19], Seurat, SPOTlight[17], TACCO, and Tangram, in determining the cell type composition of the multicell spots created based on the MOp dataset (see Methods for details on the benchmark instances). We observe that DOT outperforms other models with respect to both Jensen-Shannon and Brier Score metrics. In addition, we tested the robustness of the performance of DOT to spatial resolution and DOT's upper bound parameter on the number of cells per spot (i.e., $n$; Results). We observed that for a fixed spatial resolution, DOT exhibits a consistently high performance under different choices of parameter $n$ (Supplementary Fig. 2b). We also verified that the number of cells per spot as estimated by DOT exhibits a strong correlation with the ground truth for different choices of $n$ at different resolutions even when $n$ is smaller than the expected number of cells per spot (Supplementary Fig. 2c).

We next used single-cell level spatial data coming from osmFISH technology[30] to produce multicell data for SSp (Supplementary Notes). Subsequently, for the developing human heart, we used subcellular spatial data generated by the ISS platform[31] (Supplementary Notes). We tested the performance of DOT against the benchmark methods on these two samples, the results of which are illustrated in Fig. 3. DOT outperforms other models in the human heart sample (Fig. 3c) and is among the best-performing models in the mouse SSp sample (Fig. 3d). We also observe that DOT exhibits consistently high performance across different regions of the tissues, which implies that the performance of DOT is less sensitive to different regions/cell types of the tissue (compare to Tangram, CARD, TACCO, and Seurat in human heart, and RCTD in SSp). These results further highlight the competitive performance of DOT and its robustness in identifying the cell type composition of spots across different tissues.

## DOT recovers cortical layers in human dorsolateral prefrontal cortex
To evaluate performance on real low-resolution SRT datasets and to demonstrate the ability of DOT to transfer spatial features beyond cell types/states, we next studied transferring layer annotations in the LIBD

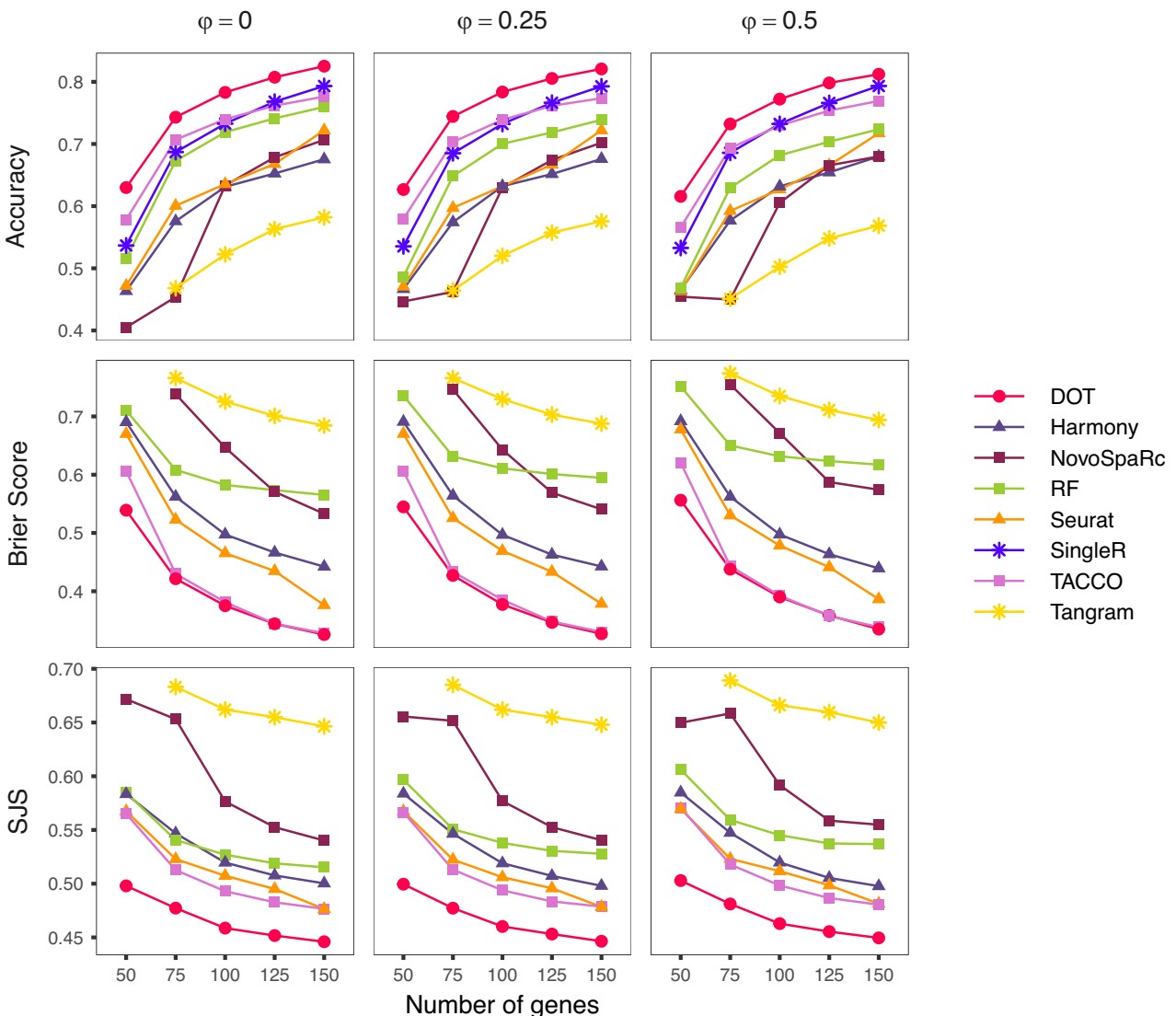

**Fig. 2 | Performance of different methods in transferring cell types to spatial locations in high-resolution spatial data.** Results are illustrated as a function of the gene coverage in the spatial data ($x$ axis) and as function of different amounts of noise in gene expression ($\varphi$). Points represent the median of 64 values. `SingleR` does not produce probabilities and is compared based on accuracy only. Source data are provided as a Source Data file.

human dorsolateral prefrontal cortex (DLPFC) dataset[32]. This dataset contains spatial gene expression profiles of 12 DLPFC samples measured with 10X Visium, with the spots manually annotated with the six layers of the human DLPFC (L1 to L6) and white matter (WM). The samples correspond to two pairs of directly adjacent serial tissue sections from three independent neurotypical adult donors (i.e., four tissue sections per donor). Here, we use DOT to transfer the layer annotations (L1 to L6, and WM) from one or a combination of reference Visium samples to a target Visium sample. We use the reference Visium samples to characterize the expression profiles of the layers (i.e., without considering the spatial information in the reference samples). Given that for each spot we know its true layer annotation, transferring the layer annotations from other Visium samples to a target Visium sample allows us to truly quantify the accuracy of DOT (and other models) in determining the layer annotation of the spots in the target sample. Moreover, based on the reference Visium samples coming from the same or different donors, we can use this dataset to assess the accuracy of the models when the reference data is matched or unmatched.

To this end, we designed a total of 36 experiments categorized into three sets, where for each target Visium sample, we created three types of reference data for transferring the layer annotations to this sample. In the first set (denoted "adjacent" in Fig. 4), for each of the 12 Visium samples, we determine the layer composition of a particular sample using its adjacent replicated Visium sample as the reference. In the second experiment (denoted "same brain" in Fig. 4), we use the three Visium samples that belong to the same donor as the reference. Finally, in the third experiment (denoted "aggregated" in Fig. 4), we use all the 11 remaining Visium samples combined as the reference.

We compared the performance of `DOT` against four top performing methods from previous experiments (i.e., `C2L`, `CARD`, `RCTD` and `TACCO`) in Fig. 4. For each experiment, we report the overall accuracy of the methods in terms of the percentage of the spots whose layers are correctly determined by each method. As illustrated in the boxplots and measured by the paired Wilcoxon signed-rank tests, `DOT` outperforms the benchmark methods with a statistically significant margin across all three experiments. In addition, while divergence from matched references (i.e., "adjacent" and "same brain") to an unmatched reference (i.e., "aggregated") affects the performance of all methods, `DOT` retains its performance above the baseline (i.e., 50% accuracy) in all 36 instances, with a median accuracy above 73% for matched reference and a median accuracy of 64% even for unmatched

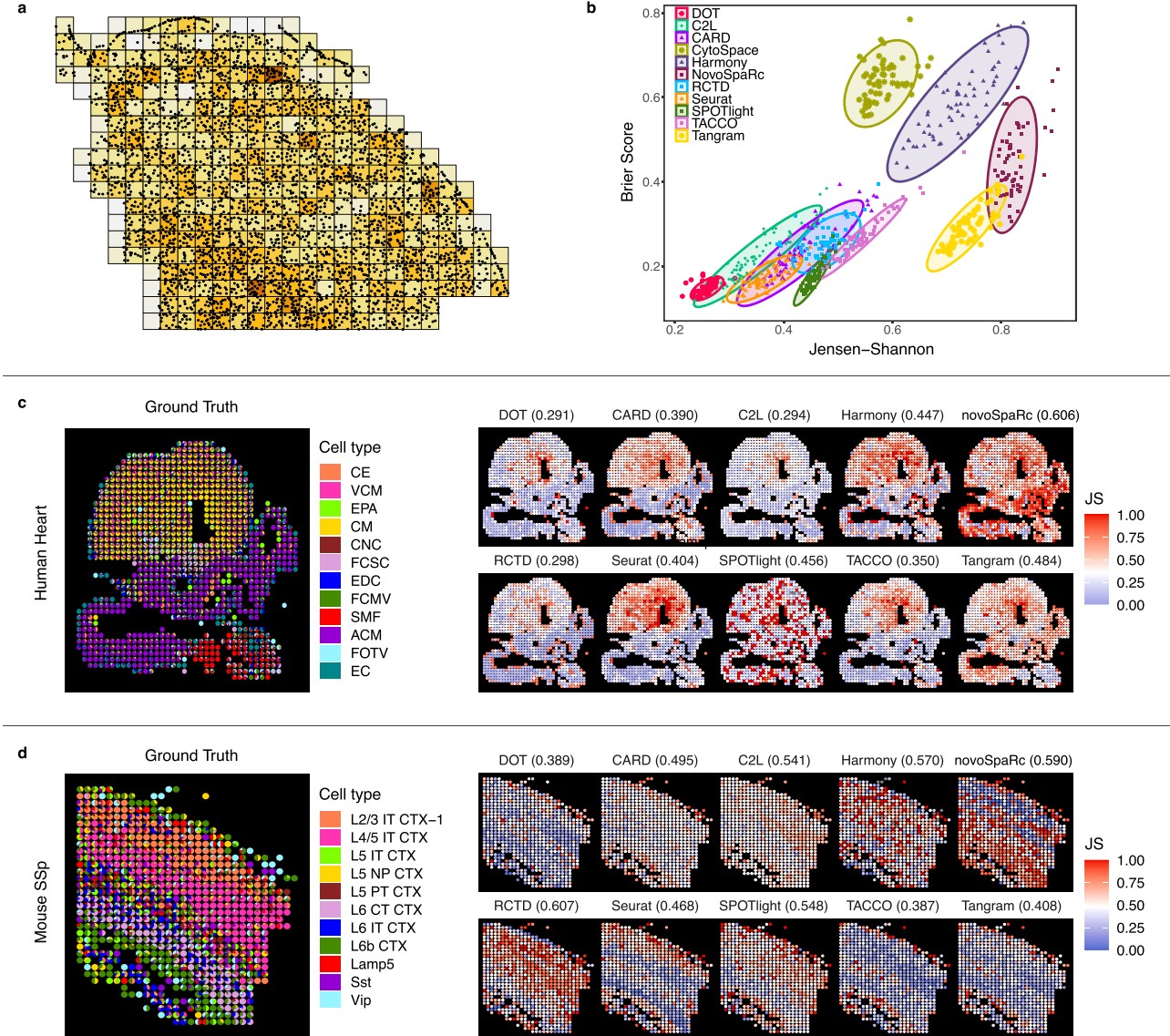

**Fig. 3 | Performance of different methods in decomposing multicell spots in low-resolution spatial data. a** Synthetic low-resolution SRT from high-resolution SRT. Dots represent cells, and tiles represent multicell spots. **b** Performance of the algorithms in the low-resolution spatial data across 64 samples of MOp with respect to Brier score and Jensen-Shannon divergence (lower better for both). Each point denotes the average performance across all spots in the sample. **c** Distribution of performance of models on each individual spot in the low-resolution spatial data of developing human heart. Each subplot shows the distribution of prediction error based on the Jensen-Shannon divergence at each spot in the spatial data, with the average value over all spots given on top of each plot (lower better). **d** Same as (**c**) for mouse SSp. Source data are provided as a Source Data file.

reference. All the while, the median accuracy for TACCO drops from 69% to 57%, and it drops to below 50% for C2L, CARD, and RCTD.

### DOT locates cell types in breast cancer in line with pathologist labels

For our second real low-resolution SRT datasets, we analyzed a total of five human breast cancer samples coming from two independent studies. The first two samples coming from ref. 33 are of the HER2+ tumor subtype and contain spatial gene expression profiles measured using Spatial Transcriptomics (ST) technology. The other three samples coming from[34] are of the triple-negative breast cancer (TNBC) tumor subtype and contain spatial gene expression profiles measured with 10X Visium technology. Both datasets contain high-level pathologist annotations (such as invasive cancer, cancer in situ, lymphocytes, immune infiltrate, and their mixtures) based on the H&E images. For both datasets, we used the scRNA-seq data coming from[34] as a

reference to infer the celltype composition of the spots in each sample based on the matched tumor subtypes, and used the pathologist annotations to validate if the cell types are enriched in the anticipated locations.

We demonstrate localization of eight major cell types in the two HER2+ ST samples in Fig. 5c. In addition, we demonstrate the size of each spot (i.e., number of cells per spot) as estimated by DOT for each sample (Fig. 5b). Similarly, we demonstrate localization of different cell types and the number of cells per spot in the three TNBC Visium samples in Fig. 6. Given that Visium offers a higher resolution (1–10 cells per spot on average) compared to ST platform (up to 200 cells per spot)[35], we set the upper bound parameter on the number of cells per spot (i.e., $n$; Results) for the ST and Visium samples to 200 and 20, respectively, and observe that DOT determines the number of cells per spot consistently with the expected density of different regions. Moreover, denser regions are enriched in smaller cells (such as

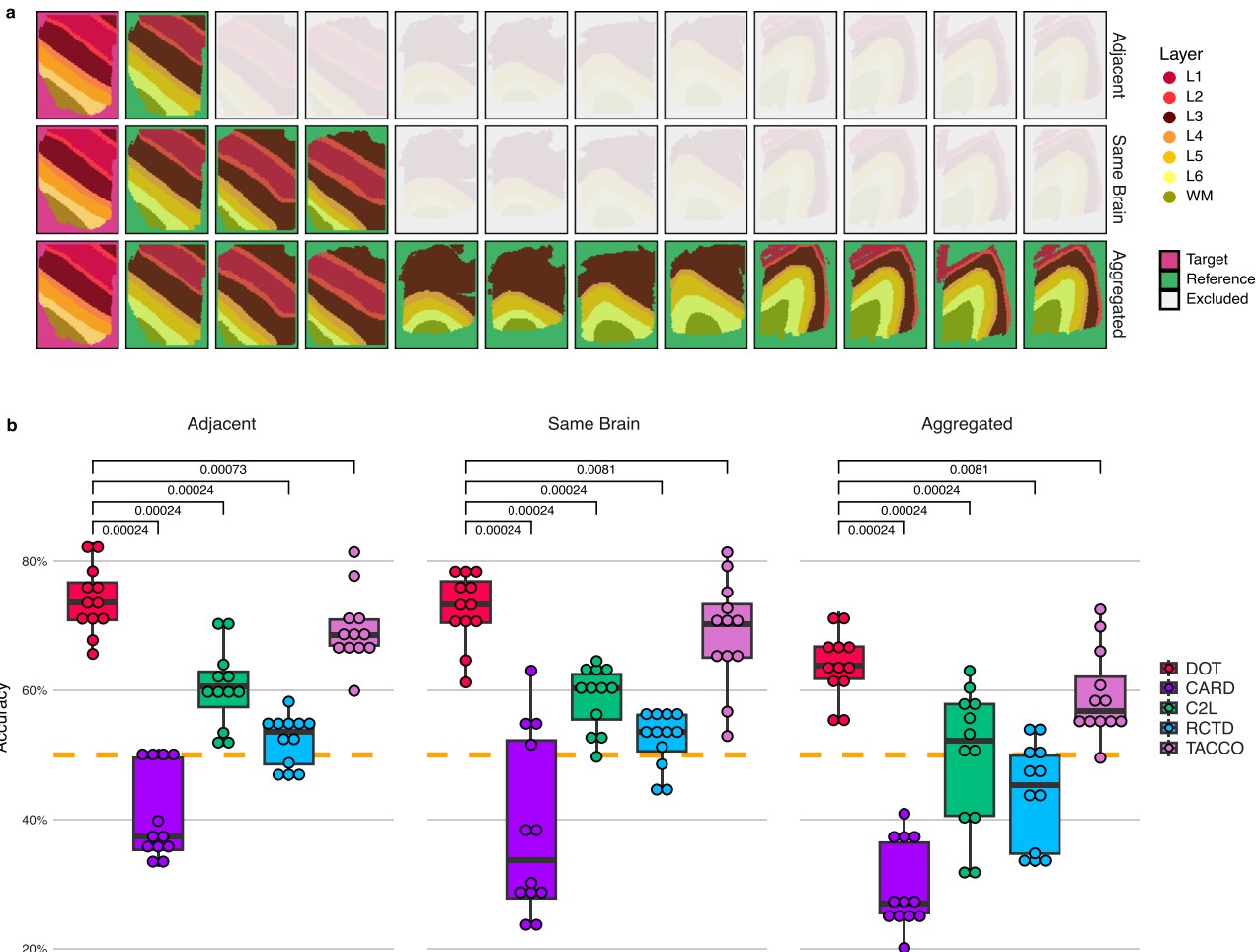

**Fig. 4 | Experiments on the LIBD data set. a** Layer annotations and experimental setup for each set of experiments. **b** Boxplots of accuracy scores of the five methods applied to the 12 DLPFC samples across three experiment settings. For each experiment set, the one-sided paired Wilcoxon signed-rank test results measure if the $n = 12$ accuracy values obtained by DOT are statistically better than the 12 respective accuracy values obtained by each model. A $p$ value of 0.00024 indicates dominance across all 12 samples. Dashed orange lines show the baseline performance (i.e., 50%). The center line in each boxplot shows the median accuracy (of 12 values), while the lower and upper hinges represent the 25th and 75th percentiles, respectively. Whiskers extend up to 1.5 × IQR (inter-quartile range) from the hinges. Source data are provided as a Source Data file.

lymphocytes/immune cells), while larger cells (such as adipose and stromal cells) appear more frequently in regions with low density.

Of note, DOT correctly localized cancerous epithelial cells in accordance with the respective pathologist annotations (e.g., invasive cancer and cancer in situ) and normal epithelial cells in accordance with normal cells (e.g., breast glands and normal glands). Moreover, DOT localizes T/B cells in accordance with lymphocytes and infiltrate immune cells and their combinations with other cells (e.g., invasive cancer), which, as expected, are also enriched in the vicinity of tumor cells[36]. In comparison, other methods tend to over- or under-estimate cancer cells, are not robust in detecting T cells, and predominantly underestimate normal epithelial cells (Fig. 5c).

**DOT estimates expression of unmeasured genes in spatial data accurately**

Given that in high-resolution SRT typically only a few genes are measured, the expression of genes that were not measured in SRT can be estimated by transferring scRNA-seq to SRT. Therefore, we evaluate the performance of DOT in estimating the expression of missing genes in the high-resolution SRT using the spatial data from breast cancer tumor microenvironment[37] (Supplementary Notes). As the high- and low-resolution SRT in this dataset come from the same tissue section,

we can use the gene expression maps in low-resolution SRT as a proxy for ground truth to evaluate the expression maps of the missing genes in the high-resolution SRT as estimated by DOT.

We started by evaluating the performance of DOT on genes that are present in the high-resolution spatial data as ground truth. In Fig. 7a we show a qualitative comparison of maps of eight genes related to breast cancer[38] produced by DOT with those of high-resolution (ground truth) and low-resolution data (approximate ground truth). The expression maps produced by DOT match the ground truth expression maps. Both DOT and the ground truth high-resolution spatial data also match the low-resolution gene expression maps almost perfectly, which further validate the quality of the solution produced by DOT. Note that due to the single-cell resolution of the high-resolution spatial data colors are brighter. Nonetheless, the spatial patterns match between all three rows.

Figure 7 b illustrates the expression maps of five genes associated with breast cancer that are not measured in the high-resolution spatial data but are estimated by DOT. For a quantitative comparison of expression maps in the high- and low-resolution SRT, given that there is no one-to-one correspondence between single-cell spots in the high-resolution and multicell spots in the low-resolution spatial data, we split the tissue into a 10 by 10 grid, and aggregated the expression of

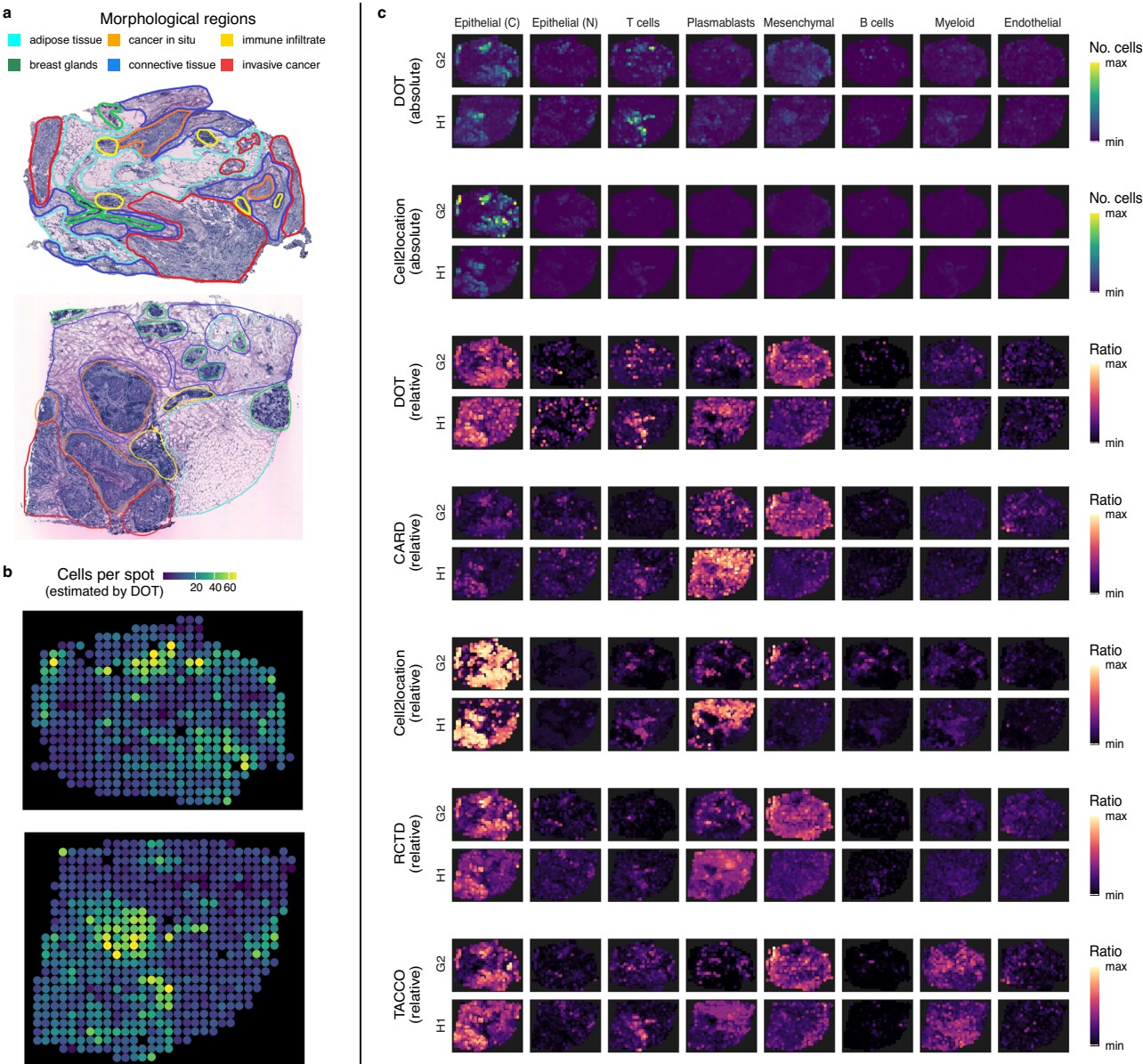

**Fig. 5 | Location and absolute abundance of cell types in two HER2+ breast tumors measured using Spatial Transcriptomics technology. a** Morphological regions as manually annotated by the pathologist into six categories: adipose tissue, breast glands, in situ cancer, connective tissue, immune infiltrate, and invasive cancer. The annotated figures are taken from ref. 33. **b** Number of cells per spot as estimated by DOT. **c** Cell type deconvolution results for the top performing methods on the HER2+ breast cancer samples. For DOT and Cell2location both absolute and relative abundances are illustrated. Epithelial cells are categorized into cancer (C) and normal (N) cells. Source data are provided as a Source Data file.

each gene within each tile. Consequently, we obtained two 100 by 18,000 matrices, one for the ground truth low-resolution spatial data and another for DOT. Figure 7c compares the column-wise cosine similarities across different genes. These results further confirm the ability of DOT in reliably estimate the expression of missing genes in high-resolution spatial data.

## Discussion

Single-cell RNA-seq and spatially resolved imaging/sequencing technologies provide each a partial picture of understanding the organization of complex tissues. To obtain a full picture, computational methods are needed to combine these two data modalities.

We present DOT, a versatile, fast and scalable optimization framework for transferring cell populations from scRNA-seq data to tissue locations, thereby transferring categorical and continuous

features from the dissociated single-cell data to the spatial data. Our optimization framework employs several alignment measures to assess the quality of transfer from different perspectives and determines the relative or absolute abundance of different cell populations in situ by combining these metrics in a multi-objective optimization model. Our metrics are designed to account for potentially different gene expression scales across the two modalities. Moreover, based on the premise that nearby locations with similar expression profiles possess similar compositions, our model leverages spatial information as well as both joint and dataset-specific genes in addition to matching the expression of common genes. In addition, whenever prior information about the abundance of cell features in the spatial data is available (e.g., estimated from a similar tissue), our model gives the user the flexibility to match these abundances to a desired level. Our model also takes into account the inherent heterogeneity of cell

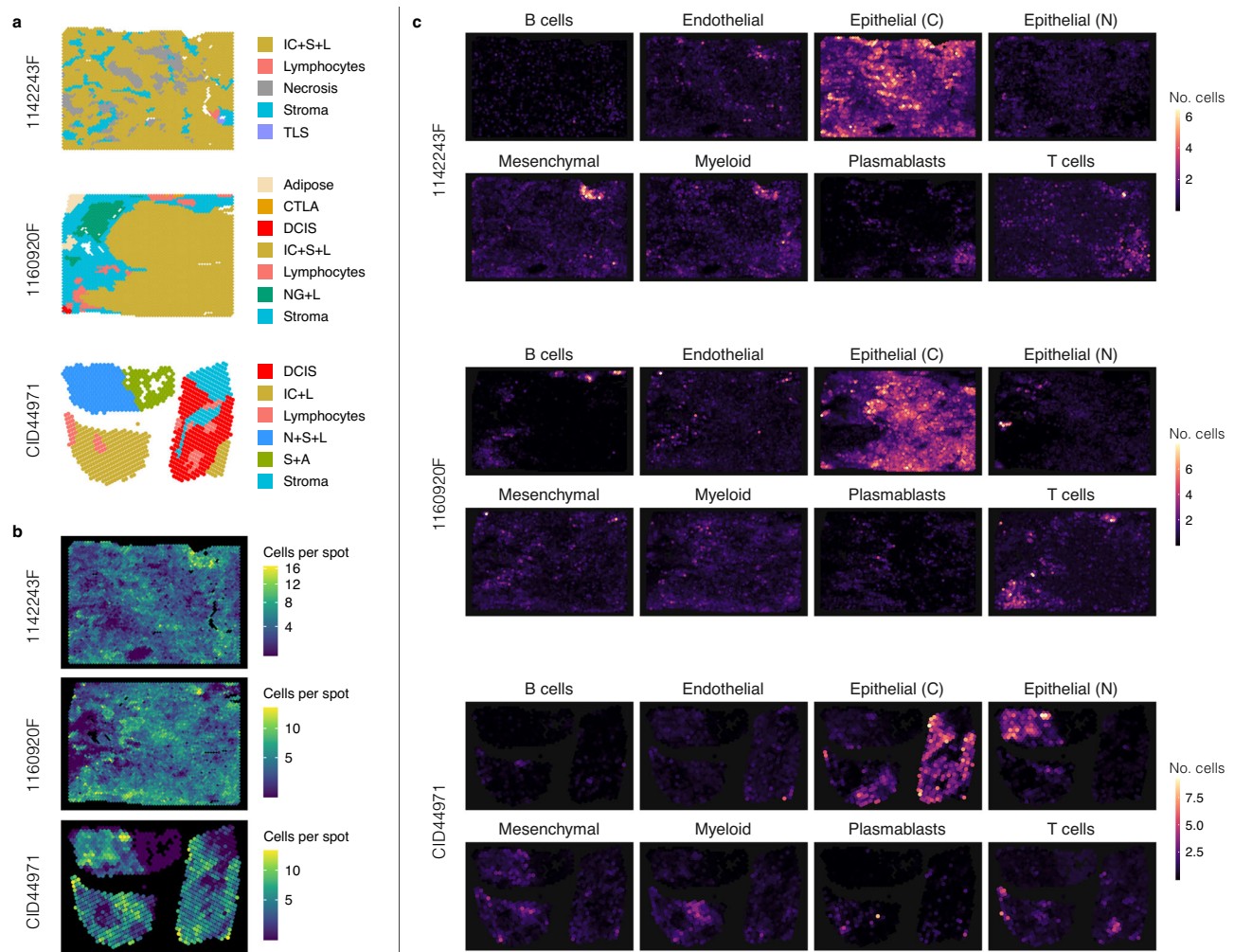

**Fig. 6 | Location and absolute abundance of cell types in three TNBC Visium samples. a** For each sample, we visualize the manual pathologist annotations. The abbreviated pathologist annotations correspond to invasive cancer (IC), stroma (S), lymphocytes (L), tertiary lymphoid structure (TLS), cancer trapped in lymphocyte aggregation (CTLA), ductal carcinoma in-situ (DCIS), and normal glands (NG). **b** Number of cells per spot as estimated by DOT. **c** Enrichment of eight major cell types at their spatial locations as estimated by DOT. Source data are provided as a Source Data file.

populations through a pre-processing step to ensure that refined subclusters of the reference are transferred.

Our model is applicable to both high-resolution (such as MER-FISH) and low-resolution (such as Visium) spatial data and can be used for gene intensity or expression count data. While we use the same optimization framework for both high- and low-resolution spatial data, our model has specific features to account for the distinct features of these modalities. In particular, our model can determine the size of spots in low-resolution spatial data and accounts for sparsity of composition of spots. For instance, in the context of inferring cell type composition of spots, this allows us to produce pure cell type compositions for high-resolution spatial data and mixed compositions for low-resolution spatial data.

While our optimization model, in its most general form, involves several terms, we have designed a solution method based on the Frank-Wolfe algorithm with special attention to scalability to large-scale dissociated and spatial data (Supplementary Figs. 3–5). Moreover, our implementation reduces the involvement of the user in parameter tuning by estimating the objective weights and other hyper parameters of the model from the data, thereby facilitating application of DOT to different problems with minimal implementation effort. Given that our model generalizes optimal transport (Supplementary Methods), we envision that DOT can be integrated with OT-based computational frameworks such as *moscot*[39] in the future.

Using experiments on data from mouse brain, human heart, and breast cancer, we showed that DOT predicts the cell type composition of spots and expression of genes in spatial data with high accuracy, often outperforming the state-of-the-art. To address the limitations of low-resolution technologies, which result in the unavailability of ground truth distributions of cell types in low-resolution spatial data, we performed objective evaluation and comparative analysis based on simulated low-resolution spatial data from high-resolution spatial data, transferring spatial features beyond cell types, and pathological annotation as a proxy for expected enrichment regions of cell types.

We established a quantifiable, objective ground truth for low-resolution data by leveraging cell-type information from MERFISH, osmFISH, and ISS across various organs and datasets and evaluated the performance of DOT by comparing it with 10 related deconvolution methods across 66 experiments. Next, we utilized manual layer annotations in the human dorsolateral prefrontal cortex dataset as ground truth. To guarantee accurate annotation transfer using DOT, we designed 36 scenarios involving annotations transferred between consecutive slides, non-consecutive slides from the same donor, and aggregated slide resources from the same organ. We assessed DOT's effectiveness in all scenarios against the four most successful methods

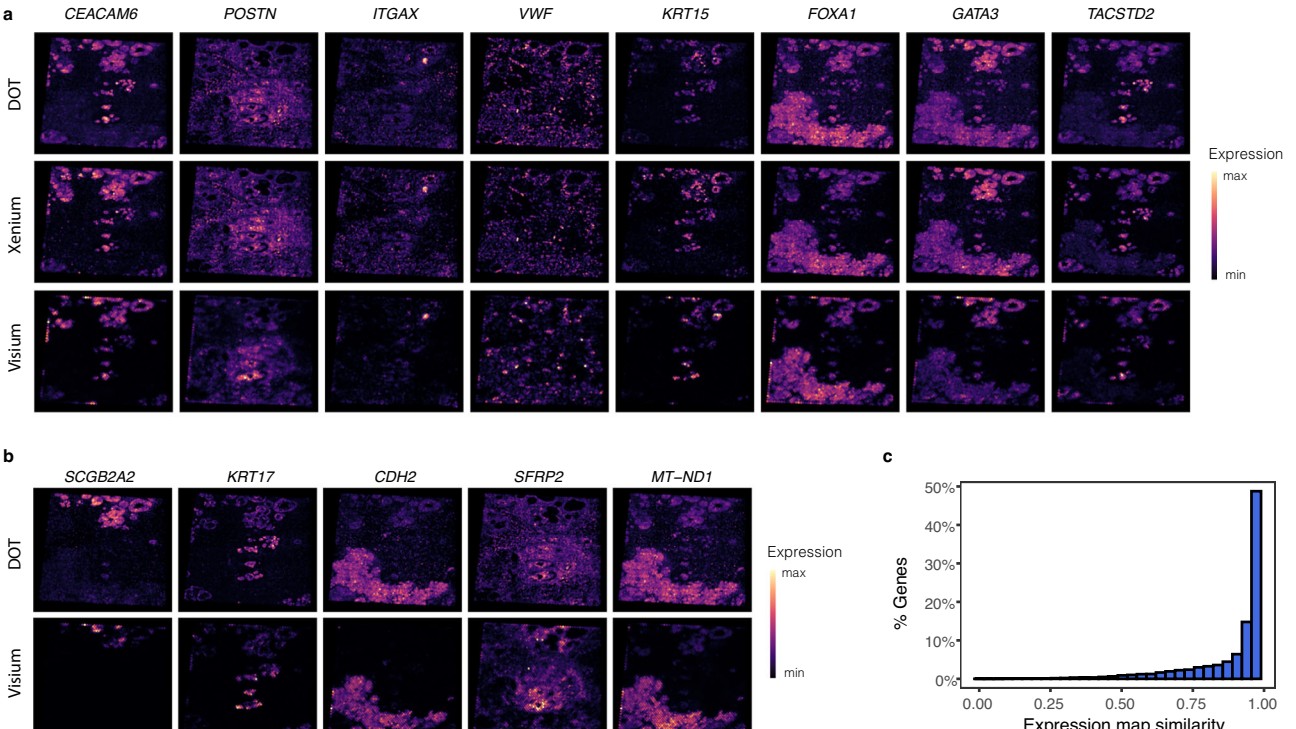

**Fig. 7 | Performance of DOT in estimating the expression of genes. a** Expression map of eight breast cancer markers as transferred from scRNA-seq to Xenium using DOT (estimated) and as measured in both Xenium (ground truth) and Visium (low-resolution proxy). **b** Expression map of five breast cancer markers that are measured in Visium but are missing in Xenium and are transferred from scRNA-seq using DOT. **c** Cosine similarity between expression maps of Visium and DOT for the genes that are not measured in Xenium. Source data are provided as a Source Data file.

from earlier evaluations, finding DOT to significantly outperform all other methods across all tasks. Lastly, we used pathological annotations to infer cell-type composition in regions across two independent breast cancer datasets (encompassing a total of five experiments). We demonstrated that DOT accurately and robustly estimates cell-type density and assigns cell types in alignment with the indirect ground truth derived from pathological annotations, offering a more detailed cell-type distribution that aligns with regional expectations.

Although we demonstrated the application of DOT in transferring cell type labels, layer annotations, and inferring the expression of missing genes, our model can be used for transferring other features such as Transcription Factor and pathway activities inferred from the reference scRNA-seq data[40]. Additionally, our optimization framework can potentially be extended to alignment of spatial multiomics by exploiting the spatial information of the different data types. As our formulation is hypothesis-free (i.e., does not rely on statistical assumptions based on mRNA counts), DOT naturally extends to applications in other omics technologies.

## Methods
### Mathematical model
From a methodological standpoint, our formulation generalizes Optimal Transport (OT) (Supplementary Methods), which is a way to match, with minimal cost, data points between two domains embedded in possibly different spaces using different variants of the Wasserstein distance[41–44]. We establish the connections between our formulation and OT formulations in Supplementary Methods, and highlight the distinct features of our model that make it more suitable for the task of transferring annotations from the reference cell populations to high- or low-resolution spatial data. Briefly, we note that our distance functions $d_i$ and $d_S$ share elements with Fused Gromov-Wasserstein (FGW)[45], which is also implemented as part of *moscot*[39].

Indeed, we present metrics for which the resulting FGW encourages similar compositions for adjacent spots with similar expression profiles, thereby its connection to our definition of set $\mathbb{P}$ and our distance function $d_S$.

Besides the specialized distance functions included in the objective function of DOT that measure the quality of the transport map from different practical perspectives, there are other substantial differences between the common components of our formulation and FGW. The first difference is that OT formulations, including FGW, construct their transportation cost matrix by assuming that each spot is assigned to exactly one cell population, discarding the fact that spots in low-resolution spatial data are composed of multiple cells coming from potentially different cell populations. In contrast, our $d_i$ distance captures both mixed and pure compositions. Moreover, the scale-invariance of $d_i$, together with our $d_c$ and $d_g$ distance functions, allow us to determine the size of spots as part of the optimization process, whereas OT variants require the sizes as given. It is also important to note that our spatial distance function $d_S$ is convex, and, by design, scales in order $O(|\mathbb{I}||\mathbb{C}|)$ (i.e., linearly in the number of spots and cell populations), while FGW formulations are non-convex and scale in $O(|\mathbb{I}|^2|\mathbb{C}| + |\mathbb{C}|^2|\mathbb{I}|)$[45], making DOT more appealing from a computational view for large-scale datasets.

**Deriving the distance functions.** To assess dissimilarity between expression vectors **a** and **b**, we introduce the distance function

$$d_{cos}(\mathbf{a}, \mathbf{b}) := \sqrt{1 - \cos(\mathbf{a}, \mathbf{b})}, \qquad (9)$$

where $\cos(\mathbf{a},\mathbf{b}) = \frac{1}{\|\mathbf{a}\|\|\mathbf{b}\|}\langle\mathbf{a},\mathbf{b}\rangle$. We note that, unlike cosine dissimilarity (i.e., $1 - \cos(\cdot,\cdot)$), $d_{cos}$ is a *metric* distance function. Moreover, $d_{cos}$ is quasi-convex for positive vectors **a** and **b**, and is scale-invariant, in the sense that it is indifferent to the magnitudes of the vectors. This is by

design, since we want to assess dissimilarity between expression vectors regardless of the measurement sensitivities of different technologies. When assessing the gene expression profiles, this also allows to measure the differences regardless of the size of spots and cell populations. We note that while $d_{cos}$ is our default choice of distance function, our optimization model naturally extends to all other scale-invariant divergences (such as LeCam divergence[46] or other symmetric $f$-divergences).

With this distance metric, by minimizing $d_i(\mathbf{Y})$ as defined in Eq. (1), we ensure that the vector of gene expressions in spot $i \in \mathbb{I}$ (i.e., $\mathbf{X}_{i,:}^S$) is most similar to the vector of gene expressions transferred to spot $i$ through $\mathbf{Y}$ (i.e., $\sum_{c \in \mathbb{C}} Y_{c,i} \mathbf{X}_{c,:}^R$). Similarly, with $d_c(\mathbf{Y})$ as defined in Eq. (2), we minimize dissimilarity between centroid of cell population $c \in \mathbb{C}$ in R (i.e., $\mathbf{X}_{c,:}^R$) and its centroid in S as determined via $\mathbf{Y}$, i.e., $\frac{1}{\rho_c} \sum_{i \in \mathbb{I}} Y_{c,i} \mathbf{X}_{i,:}^S$, where $\rho_c = \sum_{i \in \mathbb{I}} Y_{c,i}$ is the total number of spots in S assigned to $c$. Given the scale-invariance property of $d_{cos}$, we can drop $1/\rho_c$ and derive Eq. (2) as

$$d_c(\mathbf{Y}) := d_{cos}\left(\mathbf{X}_{c,:}^R, \frac{1}{\rho_c} \sum_{i \in \mathbb{I}} Y_{c,i} \mathbf{X}_{i,:}^S\right) = d_{cos}\left(\mathbf{X}_{c,:}^R, \sum_{i \in \mathbb{I}} Y_{c,i} \mathbf{X}_{i,:}^S\right).$$

We also note that $d_g(\mathbf{Y})$ as defined in Eq. (3) measures the difference between the expression map of gene $g \in \mathbb{G}$ in S (i.e., $\mathbf{X}_{:,g}^S$) and the one transferred to S through $\mathbf{Y}$ (i.e., $\sum_{c \in \mathbb{C}} Y_{c,:} X_{c,g}^R$) regardless of the scale of the expression of $g$ in S and R up to a constant multiplicative factor.

Our goal with objective (iv) as defined in Eq. (4) is to leverage the spatial information and potentially features that are contained in S but not in R to encourage spots that are adjacent in the tissue and exhibit similar expression profiles to attain similar cell type compositions. Note that we do not assume a global cell type composition similarity between all adjacent spots; instead, we employ the locally relevant spatial information. To achieve this goal, we define $\mathbb{P}$ as

$$\mathbb{P} = \left\{ (i,j) \in \mathbb{I}^2 : w_{i,j} \geq \bar{w}, \quad \| \mathbf{x}_i - \mathbf{x}_j \| \leq \bar{d}, \quad i < j \right\} \quad (10)$$

to denote the set of pairs of spots $(i, j)$ that are adjacent ($\| \mathbf{x}_i - \mathbf{x}_j \| \leq \bar{d}$) and have similar expression profiles ($w_{i,j} \geq \bar{w}$), with $\mathbf{x}_i$ denoting the spatial coordinates of spot $i$ in $\mathbb{R}^2$ or $\mathbb{R}^3$, and $w_{ij} = \cos(\mathbf{X}_{i,:}^S, \mathbf{X}_{j,:}^S)$ denoting the cosine similarity of spots $i$ and $j$ according to the full set of genes measured in S (i.e., $\mathbb{G}^S$). Here, $\bar{d}$ is a given distance threshold and $\bar{w}$ is a cutoff value for cosine similarity. As a larger $\bar{w}$ results in a smaller set $\mathbb{P}$, we can ensure that $d_S$ can be computed linearly in the number of spots $|\mathbb{I}|$ by choosing a proper value for $\bar{w}$ such that $|\mathbb{P}| = O(|\mathbb{I}|)$.

We employ Jensen-Shannon divergence defined as

$$d_{JS}(\mathbf{p}, \mathbf{q}) = \frac{1}{2} D_{KL}\left(\mathbf{p} \,\Big\|\, \frac{\mathbf{p}+\mathbf{q}}{2}\right) + \frac{1}{2} D_{KL}\left(\mathbf{q} \,\Big\|\, \frac{\mathbf{p}+\mathbf{q}}{2}\right), \quad (11)$$

to measure dissimilarity between distributions $\mathbf{q}$ and $\mathbf{p}$, where $D_{KL}(\mathbf{p} \| \mathbf{q}) = \sum_j p_j \log(p_j/q_j)$ is the Kullback-Leibler divergence[47]. We remark that $d_{JS}(\mathbf{p}, \mathbf{q})$ is strongly convex and does not require absolute continuity on distributions $\mathbf{q}$ and $\mathbf{p}$[48].

Finally, if prior information about the expected abundance of cell types in S is available (e.g., estimated from a neighboring single-cell level tissue), we denote the expected abundance of cell type $c \in \mathbb{C}$ in S by $r_c$. Note that abundance of cell type $c \in \mathbb{C}$ in S according to $\mathbf{Y}$ is $\rho_c := \sum_{i \in \mathbb{I}} Y_{c,i}$. Since $\mathbf{r}$ and $\boldsymbol{\rho}$ need not be mutually continuous, we employ $d_{JS}(\boldsymbol{\rho}, \mathbf{r})$ in Eq. (5) to measure the difference between $\mathbf{r}$ and $\boldsymbol{\rho}$.

**Cell heterogeneity.** While the cell annotations, such as cell types often correspond to distinct cell populations, significant variations may naturally exist within each cell population. This means a single vector

$\mathbf{X}_{c,:}^R$ may not properly represent the distribution of cells within cell population $c$. Consequently, transferring $c$ solely based on the centroid of cells that belong to $c$ may not capture these variations. To capture this intrinsic heterogeneity, we cluster each cell population into pre-defined $\kappa$ smaller groups using an unsupervised learning method, and produce a total of $\kappa|\mathbb{C}|$ centroids to replace the original $|\mathbb{C}|$ centroids. With this definition of centroids, we treat all terms as before, except $d_A$, since prior information about cell populations (and not their sub-clusters) is available.

Note that this approach can be extended to singleton sub-clusters, in which case DOT transfers the individual cells from the reference scRNA-seq data to the spatial data. However, transferring individual cells may be computationally expensive and prone to over-fitting, particularly when the reference data and the spatial data are not matched or when there is a significant drop-out in the reference scRNA-seq data. In general, we treat the sub-clusters with very few cells as outliers and remove them to obtain a set $\mathbb{K}_c$ of sub-clusters for cell population $c \in \mathbb{C}$. Once $\mathbf{Y}$ is obtained, $\sum_{k \in \mathbb{K}_c} Y_{k,i}$ determines the abundance of cell population $c$ in spot $i$.

**Sparsity of composition.** As previously discussed, spatial data are either high-resolution (single-cell level) or low-resolution (multi-cell level). In the case of high-resolution spatial data, given that each spot corresponds to an individual cell (i.e., $n_i = 1$), we expect that spots are pure (as opposed to mixed), in the sense that we prefer $Y_{c,i}$ close to 0 or 1. In general, assuming that size of spot $i$ is $\bar{n}_i$ (i.e., $\bar{n}_i = \sum_{c \in \mathbb{C}} Y_{c,i}$) and $Y_{c,i} \in \{0, \bar{n}_i\}$, then $Y_{c,i} = \bar{n}_i$ for exactly one category $c$ and is zero for all other categories. Consequently, for binary-valued $\mathbf{Y}$ we obtain

$$d_{cos}\left(\mathbf{X}_{i,:}^S, \sum_{c \in \mathbb{C}} Y_{c,i} \mathbf{X}_{c,:}^R\right) = \frac{1}{\bar{n}_i} \sum_{c \in \mathbb{C}} Y_{c,i} d_{cos}\left(\mathbf{X}_{i,:}^S, \mathbf{X}_{c,:}^R\right),$$

which is linear in $\mathbf{Y}$ for fixed $\bar{n}_i$. As linear objectives promote sparse (or corner point) solutions, we may control the level of sparsity of the solution by introducing a parameter $\theta \in [0, 1]$ and redefining $d_i(\mathbf{Y})$ as

$$d_i(\mathbf{Y}) = (1 - \theta) d_{cos}\left(\mathbf{X}_{i,:}^S, \sum_{c \in \mathbb{C}} Y_{c,i} \mathbf{X}_{c,:}^R\right) + \frac{\theta}{\bar{n}_i} \sum_{c \in \mathbb{C}} Y_{c,i} d_{cos}\left(\mathbf{X}_{i,:}^S, \mathbf{X}_{c,:}^R\right). \quad (12)$$

Note that a higher value for $\theta$ yields a sparser solution. Indeed, with $\theta = 1$ and zero weights assigned to other objectives, the optimal solution will be completely binary. Note that $\bar{n}_i$ acts as a penalty weight and can be set to a fixed value (e.g., $n_i$).

**A fast Frank-Wolfe implementation**
We propose a solution to the DOT model based on the Frank-Wolfe (FW) algorithm[49,50], which is a first-order method for solving non-linear optimization problems of the form $\min_{\mathbf{x} \in \mathbb{X}} f(\mathbf{x})$, where $f : \mathbb{R}^n \rightarrow \mathbb{R}$ is a (potentially non-convex) continuously differentiable function over the convex and compact set $\mathbb{X}$. FW operates by replacing the non-linear objective function d with its linear approximation $\tilde{f}(\mathbf{x}) := f(\mathbf{x}^{(0)}) + \nabla_{\mathbf{x}} f(\mathbf{x}^{(0)})^\top (\mathbf{x} - \mathbf{x}^{(0)})$ at a trial point $\mathbf{x}^{(0)} \in \mathbb{X}$, and solving a simpler problem $\hat{\mathbf{x}} := \arg\min_{\mathbf{x} \in \mathbb{X}} \tilde{f}(\mathbf{x})$ to produce an "atom" solution $\hat{\mathbf{x}}$. The algorithm then iterates by taking a convex combination of $\mathbf{x}^{(0)}$ and $\hat{\mathbf{x}}$ to produce the next trial point $\mathbf{x}^{(1)}$, which remains feasible thanks to convexity of $\mathbb{X}$. The FW algorithm is described in Algorithm 1, in which $f(\mathbf{Y})$ is the objective function in Eq. (6). Implementation details can be found in Supplementary Methods.

**Algorithm 1.** Frank-Wolfe algorithm for DOT

1  Set $t = 0$; find an initial solution $\mathbf{Y}^{(0)}$
2  **while** *not converged* **do**
3 Compute gradient $\mathbf{\Delta}^{(t)} = \nabla_{\mathbf{Y}} f(\mathbf{Y}^{(t)})$
4 Compute the atom solution $\hat{\mathbf{Y}}^{(t)}$:
5 **for** *each spot* $i \in \mathbb{I}$ **do**
6 Find the current best category $\hat{c} = \arg\min_{c \in \mathbb{C}}\{\Delta^{(t)}_{c,i}\}$.
7 Set $\hat{Y}^{(t)}_{c,i} = 0$ for $c \neq \hat{c}$.
8 If $\Delta^{(t)}_{c,i} < 0$, set $\hat{Y}^{(t)}_{\hat{c},i} = n_i$, otherwise set $\hat{Y}^{(t)}_{\hat{c},i} = 1$
9 Update $\mathbf{Y}^{(t+1)} = \mathbf{Y}^{(t)} + \frac{2}{2+t}(\hat{\mathbf{Y}}^{(t)} - \mathbf{Y}^{(t)})$
10 $t \leftarrow t + 1$

While the DOT model is not separable, its linear approximation can be decomposed to $|\mathbb{I}|$ independent subproblems, one for each spot $i \in \mathbb{I}$. This is because, unlike conventional OT formulations, we do not require the marginal distribution of cell populations (i.e., $\sum_{i \in \mathbb{I}} Y_{c,i}$) to be equal to their expected distribution (i.e., $r_c$), but have penalized their deviations in the objective function using $d_A$ defined in Eq. (5). The subproblem $i$ then becomes

$$\min\left\{\langle \mathbf{Y}_{:,i} \mathbf{\Delta}^{(t)}_{:,i} \rangle : \mathbf{Y}_{:,i} \in \mathbb{R}^{|\mathbb{C}|}_{+}, \quad 1 \leq \sum_{c \in \mathbb{C}} Y_{c,i} \leq n_i\right\}$$

which has a simple solution. Denoting the category with smallest coefficient by $\hat{c}$, if cost coefficient of $\hat{c}$ is negative then $Y_{\hat{c},i} = n_i$, otherwise $Y_{\hat{c},i} = 1$. Consequently, $Y_{c,i} = 0$ for all other categories. This property of Algorithm 1 enables it to efficiently tackle problems with a large number of spots in the spatial data.

## Experimental setup

**Parameter setting.** In its most general form, our multi-objective formulation for DOT involves the penalty weights $\lambda_C$, $\lambda_G$, $\lambda_S$ and $\lambda_A$ in Eq. (6), the upper bound on size of spots $n$ in Eq. (8), and the spatial neighborhood parameters $\bar{w}$ and $\bar{r}$ that derive the definition of spatial pairs $\mathbb{P}$ in Eq. (10). Here, we show how all of these parameters can be inferred from the data, hence eliminating the need for the user to tune these parameters.

We set the penalty weights in such a way that all objectives contribute roughly equally to the objective function. More specifically, we choose $\lambda_C \propto \frac{|\mathbb{I}|}{|\mathbb{C}|}$ and $\lambda_G \propto \frac{|\mathbb{I}|}{|\mathbb{G}|}$ since $\sum_{i \in \mathbb{I}} d_i(\mathbf{Y})$ is in the range of 0 and $|\mathbb{I}|$, while $\sum_{c \in \mathbb{C}} d_c(\mathbf{Y})$ and $\sum_{g \in \mathbb{G}} d_g(\mathbf{Y})$ are upperbounded by $|\mathbb{C}|$ and $|\mathbb{G}|$, respectively. More precisely, we set $\lambda_C = 0.5 \frac{|\mathbb{I}|}{|\mathbb{C}|}$ and $\lambda_G = 1.25 \frac{|\mathbb{I}|}{|\mathbb{G}|}$ for high-resolution SRT, and set $\lambda_C = 0.6 \frac{|\mathbb{I}|}{|\mathbb{C}|}$ and $\lambda_G = 1.5 \frac{|\mathbb{I}|}{|\mathbb{G}|}$ for low-resolution SRT (see Supplementary Fig. 6). We set the upper bound on the size of spots to $n = 1$ in high-resolution SRT. For low-resolution SRT, we set $n$ according to the expected resolution of the technology. For instance, for 10X Visium we set the upper bound parameter to $n = 20$ to accommodate 1–10 cells per spot on average[51], while for Spatial Transcriptomics (ST), we set $n = 200$ as spots in ST platform contain up to 200 cells per sample[35]. If the parameter is not specified, we set $n = \frac{N}{|\mathbb{I}|}$ where $N$ is the total number of cells that can fit the spatial data, where we employ a generalized linear regression model to estimate $N$ (Supplementary Methods). We also choose $\lambda_S \propto \frac{|\mathbb{I}|}{n|\mathbb{P}|}$ as it is not difficult to verify that $0 \leq d_S(\mathbf{Y}) \leq n|\mathbb{P}|$ when Jensen-Shannon divergence is computed in base 2 logarithm. More precisely, we set $\lambda_S = 0.5 \frac{|\mathbb{I}|}{|\mathbb{P}|}$ and $\lambda_S = 0.4 \frac{|\mathbb{I}|}{n|\mathbb{P}|}$ for high- and low-resolution SRT, respectively (Supplementary Fig. 6). Similarly, whenever prior information about the expected abundance of cell populations (i.e., $\mathbf{r}$) is available, we scale $\mathbf{r}$ such that $\sum_{c \in \mathbb{C}} r_c \approx N$ and set $\lambda_A = \frac{|\mathbb{I}|}{N} = \frac{1}{n}$. When such information is not available, we turn off this objective by setting $\lambda_A = 0$.

We set the sparsity parameter $\theta = 0.6$ for high-resolution SRT, and set $\theta = 0.4$ for low-resolution SRT (Supplementary Fig. 6). To capture heterogeneity of cell populations, we clustered each cell population $c \in \mathbb{C}$ into $\kappa = 10$ clusters and filtered out the sub-clusters containing less than 1% of the total number of cells in $c$. To compute the distance threshold $\bar{d}$, we computed the Euclidean distance of each spot to its 8 closest spots in space to mimic the number of adjacent tiles in a 2D regular grid, yielding $8|\mathbb{I}|$ values. We then took $\bar{d}$ as the 90th percentile of these values. Finally, we set $\bar{w}$ to the maximum of 0.6 and the largest value that maintains $|\mathbb{P}| \leq |\mathbb{I}|$ to ensure meaningful spatial neighborhoods and that $d_S$ scales linearly in the number of spots for the sake of computational efficiency.

For `RCTD`, `SPOTlight`, `Tangram`, `C2L`, and `NovoSpaRc`, we used the default parameters suggested by the authors with the following exceptions. For `RCTD` we set the parameter `UMI_min` to 50 to prevent the model from removing too many cells from the data. Given the large number of cell types in the mouse MOp datasets, for `SPOTlight` we reduced the number of cells per cell type to 100 to enhance the computation time. Similarly, as `Tangram` was not able to produce results in a reasonable time for the MOp instances, we randomly selected 500 cells per cell type to reduce the computation time. For `C2L`, we used 20000 epochs to balance computation performance and accuracy. We reduced the number of epochs to 2000 in the LIDB dataset to reduce the running time for `C2L` down to 12 hours per sample. For `NovoSpaRc` we set the fusion parameter $\alpha$ to 0.5 to use both expression and spatial information, and down-sampled the reference data to at most 500 cells per cell type to improve computational efficiency. For `Seurat`, `Harmony`, `SingleR`, `TACCO`, and `CytoSPACE`, we followed the package documentations, with functions used with default parameters. For `RF` we used the implementation provided in the R package `ranger`[52] with all parameters set at their default values.

**Performance metrics.** We used three metrics for comparing the performance of different models in predicting the composition of spots. In our high-resolution spatial data coming from the MOp region of mouse brain, we know the cell type of each single-cell spot given as $P_{c,i} = 1$ if spot $i$ is of type $c$, and $P_{c,i} = 0$ otherwise. We can therefore treat the cell type prediction as a multiclass classification task.

*Accuracy* is the proportion of correctly classified spots (i.e., sum of the main diagonal in the confusion matrix) over all spots. We also use *Brier Score*, also known as mean squared error, to compare the accuracy of membership probabilities produced by each model:

$$\text{Brier Score} = |\mathbb{I}|^{-1} \sum_{i \in \mathbb{I}} \sum_{c \in \mathbb{C}} (Y_{c,i} - P_{c,i})^2,$$

where $Y_{c,i}$ is the probability predicted by the model that spot $i$ is of cell type $c$. As Brier Score is a strictly proper scoring rule for measuring the accuracy of probabilistic predictions[53], lower Brier Score implies better-calibrated probabilities.

Besides the cell type that each spot is annotated with, we can produce a cell type probability distribution for each spot by considering the cell type of its neighboring spots, using a Gaussian smoothing kernel of the form

$$\tilde{P}_{c,i} = \left(\sum_{j \in \mathbb{I}} K_{i,j}\right)^{-1} \sum_{j \in \mathbb{I}} K_{i,j} P_{c,j},$$

where $K_{i,j} = \exp\left(-\|\mathbf{x}_i - \mathbf{x}_j\|^2 / 2\sigma^2\right)$ and $\sigma$ is the kernel width parameter which we set to $0.5\bar{d}$. Note that as spot $j$ becomes closer to spot $i$, its label contributes more to the probability distribution at spot $i$.

Using these probabilities, we also introduce the *Spatial Jensen-Shannon* divergence to compare the probability distributions assigned to spots (i.e., **Y**) with the smoothed probabilities (i.e., $\tilde{\mathbf{P}}$)

$$\mathrm{SJS} = \frac{1}{|\mathbb{I}|}\sum_{i \in \mathbb{I}} d_{\mathrm{JS}}(\mathbf{Y}_{:,i}, \tilde{\mathbf{P}}_{:,i}),$$

where $d_{\mathrm{JS}}(\mathbf{Y}_{:,i}, \tilde{\mathbf{P}}_{:,i})$ is the Jensen-Shannon divergence between probability distributions $\mathbf{Y}_{:,i}$ and $\tilde{\mathbf{P}}_{:,i}$ with base 2 logarithm as defined in Eq. (11).

Unlike the high-resolution spatial data, the ground truth $P_{c,i}$ in the low-resolution spatial data corresponds to relative abundance of cell type $c$ in spot $i$. We can therefore assess the performance of each model by comparing the probability distributions $\mathbf{P}_{:,i}$ and the estimated probabilities (i.e., $\mathbf{Y}_{:,i}$) using Brier Score or Jensen-Shannon metrics.

**Data preparation.** For experiments on transferring cell types to high-resolution spatial data (Results), with each sample of the MERFISH MOp (Supplementary Notes), we created a reference single-cell data using all the 280,186 cells, except the cells contained in the sample, and the 254 genes to estimate the centroids of the 99 reference cell types. We further created 15 high-resolution spatial datasets for each sample (i.e., a total of 960 spatial datasets) as follows. To simulate the effect of number of shared features between the spatial and scRNA-seq data, we assumed that only a subset of the 254 genes are available in the spatial data by selecting the first $|\mathbb{G}|$ genes, where $|\mathbb{G}| \in \{50,75,100,125,150\}$ (i.e., 20%, 30%, 40%, 50%, 60% of genes). Moreover, to simulate the effect of differences in measurement sensitivities of different technologies, we introduced random noise in the spatial data by multiplying the expression of gene $g$ in spot $i$ by $1 + \beta_{i,g}$, where $\beta_{i,g} \sim U(-\varphi, \varphi)$ with $\varphi \in \{0, 0.25, 0.5\}$.

We produced low-resolution transcriptome-wide SRT samples based on the MERFISH MOp and scRNA-seq MOp[54] (Supplementary Notes) as follows. Instead of placing cells from scRNA-seq MOp in random spatial locations, we used the MERFISH MOp data to guide the anticipated location of different cell types and assign cells from the scRNA-seq data to spatial locations based on the common "subclass" annotations between these two datasets. More specifically, for each of the 64 MERFISH MOp samples, we replaced each cell in the MERFISH MOp data with a randomly selected cell in the scRNA-seq MOp data of the same subclass. We then lowered the resolution of spatial data by splitting each sample into regular grids of length $100\mu m$. Finally, we aggregated the expression profiles of cells within each tile to produce transcriptome-wide spots.

For experiments on estimating the expression of unmeasured genes in low-coverage spatial data (Results), we matched the common capture areas of high- and low-resolution spatial data using the Hematoxylin-Eosin (H&E) images accompanying these spatial data (Supplementary Fig. 7), which corresponded to 134,664 cells in the high-resolution and 3928 spots in the low-resolution spatial data. Given that the task at hand is to estimate the expression of missing genes in the high-resolution spatial data, we performed community detection on the graph of shared nearest neighbors of cells in scRNA-seq using the Leiden implementation in[27], which is common practice in single-cell analysis and is used as a first step towards cell population identification (note that the reference scRNA-seq does not contain cell type annotations). This resulted in 218 clusters; we then transferred the centroids of these clusters to the high-resolution spatial data. (We also tried as high as 1000 fine-grained clusters but got essentially the same results).

**Statistics and reproducibility**
Statistical tests used in data analysis are listed in figure legends and/or relevant sections in Methods. No statistical method was used to predetermine the sample size. No data were excluded from the analyses. The experiments were not randomized.

**Reporting summary**
Further information on research design is available in the Nature Portfolio Reporting Summary linked to this article.

## Data availability
Publicly available single-cell RNA-seq and spatial data can be accessed via the following accession numbers or the links provided. MERFISH data of mouse MOp[24] can be accessed at the Brain Image Library: https://doi.org/10.35077/g.21. Single-cell RNA-seq data of mouse MOp[54] and SSp[55] can be accessed at the NeMO Archive for the BRAIN Initiative Cell Census Network via https://assets.nemoarchive.org/dat-ch1nqb7and https://assets.nemoarchive.org/dat-jb2f34y, respectively. osmFISH data of mouse SSp is available at http://linnarssonlab.org/osmFISH/. ISS and scRNA-seq data of the developing human heart[31] is available at the European Genome-phenome Archive via accession number EGAS00001003996. The LIBD human dorsolateral prefrontal cortex (DLPFC) spatial transcriptomics data can be accessed via https://research.libd.org/spatialLIBD. The human breast cancer scRNA-seq and Visium samples[34] can be accessed via GSE176078 and https://doi.org/10.5281/zenodo.4739739, respectively. The HER2+ breast tumors measured by Spatial Transcriptomics[33] are available at https://doi.org/10.5281/zenodo.4751624. Xenium, Visium and scRNA-seq data of human breast cancer[37] can be accessed at https://www.10xgenomics.com/products/xenium-in-situ/preview-dataset-human-breast. More detailed description of these datasets can be found in Supplementary Notes. Source data are provided as a Source Data file. Source data are provided with this paper.

## Code availability
DOT is available as the open-source R package DOT, with source code freely available at https://github.com/saezlab/dot[56]. Scripts required for reproducing the results are available at https://github.com/saezlab/dot_experiments.

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

## Acknowledgements

We thank Ricardo O. Ramirez-Flores (Heidelberg University) and Zeinab Mokhtari (GSK) for their valuable discussions.

## Author contributions

A.R., J.S.R. and M.F.S. conceptualized the project. A.R. developed the methodology and algorithm, designed the experiments, performed the comparisons, evaluations, and visualizations presented in this work, and wrote the original draft. A.R. developed the software with the support of J.T. L.V.S. and M.F.S. helped shape the work presented here and contributed to the discussion of the results. J.T. and J.S.R. contributed to editing and review. J.S.R. and J.T. supervised the project with the help of M.F.S. All authors consent to publication of this article.

## Funding

## Competing interests

A.R. is supported by funding from GSK. J.S.R. reports funding from GSK, Pfizer and Sanofi and fees/honoraria from Travere Therapeutics, Stadapharm, Astex, Pfizer and Grunenthal. The remaining authors declare no competing interests.
