## [Peer Review File · Nature Communications]

DOT: A flexible multi-objective optimization framework for transferring features across single-cell and spatial omicsReviewer #1 (Remarks to the Author):

In the manuscript entitled "DOT: A flexible multi-objective optimization framework for transferring features across single-cell and spatial omics", Rahimi et al. developed a multi-objective optimization framework, DOT, to transfer cellular features across scRNA-seq and SRT data. Once the "transfer matrix" is obtained, DOT can perform various downstream tasks in spatial transcriptomic data analysis, such as cell type deconvolution and gene imputation. The authors benchmarked DOT and other methods on three SRT datasets and demonstrated the good performance as well as scalability of their method. In my opinion, the Method section of this manuscript is detailed, and the discussion on the connections between the formulation of DOT and standard OT formulations is also quite interesting. However, my main concern is the novelty of the manuscript and the comprehensiveness of validation.

My detailed comments are listed below and I hoped they would be helpful for the authors to further improve the manuscript.

Major

1. Since a lot of cell type deconvolution or gene imputation methods have been published so far, I think the authors are supposed to further highlight the unique advantages or novelty of DOT compared with other methods.
2. The authors said, "Recent methods mostly rely on..., do not use the spatial relationships between cells in the spatial data". However, a recently published method CARD (Ma et al., Nature Biotechnology, 2022) has utilized the spatial correlation structure in the spatial transcriptomics data for cell type deconvolution. The authors should revise this paragraph and compare DOT with CARD.
3. Other cell type deconvolution methods are also recommended for comparison:
 - CytoSPACE (Vahid et al., Nature Biotechnology, 2023)
 - TACCO (Mages et al., Nature Biotechnology, 2023)
4. The authors should benchmark DOT's computational requirements (including runtime and memory requirements) with a range of dataset sizes, rather than simply stating the results based on only four sets of data.
5. Most of the datasets used for benchmarking are generated by imaging-based techniques (MERFISH, osmFISH, and ISS) and contain only tens to hundreds of genes. I'm curious how DOT performs on datasets with more genes. To achieve this, the authors could utilize simulated SRT data constructed from scRNA-seq data (as implemented in the Cell2location) or constructed from Slide-seq data (as implemented in the CytoSPACE).
6. The authors should benchmark DOT's robustness on datasets with a range of spot resolutions (for example, mean of 5, 10, 20, ... cells per spot).
7. I only saw the application of DOT on gene imputation. I suggest the authors provide some cell type deconvolution applications of DOT on real low-resolution SRT data (such as ST, 10x Visium, or Slide-seq).

Minor

8. The description of "Sub population" is confusing. I suggest the authors revise it to "Cell population".

Reviewer #2 (Remarks to the Author):

The authors proposed a novel tool, DOT, for integrating single-cell and spatial transcriptome in this manuscript. DOT attempts to match the gene expressions between single cell and spatial transcriptome, cell subpopulation abundance and capture spatial relations by a multi-objective optimization model. The idea of DOT is interesting but the manuscript is not well written and the evaluation of the performance is also not well designed and the results are not convincing for me.

1. In "DOT locates cell types in high-resolution spatial data" section, DOT is compared with RCTD, Seurat, Tangram and SingleR. RCTD is designed for cell type decomposition of spatial transcriptomics such as slide-seq and Visium, not for the image-based method such as MERFISH

which contains only hundreds of genes. RCTD should not be included in this scenario. In addition, some popular integration and transferring methods such as harmony, TACCO and Spatial-ID should be included.

2. In "DOT determines cell type abundances in low-resolution spatial data" section, the authors evaluated the performance of decomposition of spatial transcriptomics. Decomposition is only used for low resolution low-resolution SRT such as Visium which includes the whole transcriptome. But the datasets used in the evaluation are produced from MERFISH and osmFISH which include only 254 genes and 33 genes, respectively. So these two datasets are totally NOT suitable for the decomposition. The results are meaningless. Some other deconvolution methods such as CARD should be included here as well.

3. DOT is a multiple objective optimization model which have several weights parameters for objectives. How do these user-defined penalty weights effect the performance?

4. The Methods section is too long. Section 4.1 "related work" should be moved to Introduction and some mathematical details should be moved to supplementary.

Detailed response to reviews on “*DOT: A flexible multi-objective optimization framework for transferring features across single-cell and spatial omics*”

Manuscript ID: NCOMMS-23-36424-T

We would like to thank the reviewers and the associate editor for their constructive comments. Below, we provide their comments in normal typeface and our responses as highlighted.

Reviewer #1 (Remarks to the Author):

In the manuscript entitled “DOT: A flexible multi-objective optimization framework for transferring features across single-cell and spatial omics”, Rahimi et al. developed a multi-objective optimization framework, DOT, to transfer cellular features across scRNA-seq and SRT data. Once the “transfer matrix” is obtained, DOT can perform various downstream tasks in spatial transcriptomic data analysis, such as cell type deconvolution and gene imputation. The authors benchmarked DOT and other methods on three SRT datasets and demonstrated the good performance as well as scalability of their method. In my opinion, the Method section of this manuscript is detailed, and the discussion on the connections between the formulation of DOT and standard OT formulations is also quite interesting. However, my main concern is the novelty of the manuscript and the comprehensiveness of validation.

My detailed comments are listed below and I hoped they would be helpful for the authors to further improve the manuscript.

Comment #1: Since a lot of cell type deconvolution or gene imputation methods have been published so far, I think the authors are supposed to further highlight the unique advantages or novelty of DOT compared with other methods.

Thank you for your helpful comment. We have now revised the manuscript to highlight the unique features of DOT compared with other methods. In particular,

we have moved parts of the methods subsection discussing the related work to the Introduction and, as stated below, we more explicitly stated the limitations that DOT addresses, which in turn set it apart from other methods:

“In addition to being applicable to various types of spatial omics, a distinctive feature of DOT is that it exploits the spatial context and the genes that are present in scRNA-seq or SRT but missing in the other. This is in contrast to approaches that do not use the spatial localization information in the spatial omics and rely on the genes that are mutually captured by both scRNA-seq and SRT without using the remaining genes exclusively captured in each modality. One major challenge that we address with DOT is taking into consideration the spatial context of the data explicitly. On the one hand, neglecting the spatial context is equivalent to assuming random placement of spots in the space, which is at odds with the established structure-function relationship of tissues. On the other hand, assuming a uniform dependence between cell type composition and spatial location across different regions of the tissue (as assumed, e.g., in CARD) might not hold for complex tissues. In contrast, DOT leverages the spatial context in a local manner without assuming a global correlation. Our local view of spatial features allows us to utilize them only when it is beneficial to do so, based on a threshold on the similarity of gene expression of adjacent spots. Another feature of DOT is that it exploits the genes that are not mutually measured in scRNA-seq and SRT. Considering only a subset of genes limits the applicability of these methods to cases where the two data sets share several informative genes, which might not be the case when different technologies are used for profiling, or when few genes are measured in the spatial data (e.g., in MERFISH). In DOT, we use the genes exclusively measured in scRNA-seq to refine the cell populations and capture their heterogeneity by sub-clustering the cell populations into refined clusters, and use the distinct genes in SRT to inform the locally relevant spatial structures.

Another distinctive feature of DOT is that it is applicable to both high- and low-resolution SRT, as our model is capable of inferring membership probabilities for the former and absolute abundance of cell populations and size of spots in the latter. Additionally, DOT works for both absolute counts and normalized expressions. This distinguishes our model from optimal transport-based models (such as TACCO) and deep learning methods (such as Tangram), which do not offer absolute abundances in low-resolution, and statistical methods (such as cell2location and RCTD), which rely on absolute mRNA counts.”

Comment #2: The authors said, “Recent methods mostly rely on..., do not use the spatial relationships between cells in the spatial data”. However, a recently published method CARD (Ma et al., Nature Biotechnology, 2022) has utilized the spatial correlation structure in the spatial transcriptomics data for cell type deconvolution. The authors should revise this paragraph and compare DOT with CARD.

Thank you for bringing this to our attention. We were not aware of CARD at the time of writing the first draft of the paper. We now include CARD in our overview of related work and benchmark (subsections 2.2 and 2.4). We discuss in more detail the differences in the use of the spatial context (please see above) and quantify the differences in performance between CARD and DOT (shown in Figures 2, 3 and 4). For your reference, we have included these results in Figures R1, R2, R3 and R4 with proper references to the figures in the manuscript.

Figure R1 (Figure 2 in the manuscript). Performance of different methods in transferring cell types to spatial locations in high-resolution spatial data as function of the gene coverage in the spatial data and as function of different amounts of noise in gene expression.

Figure R2 (Figure 3b in the manuscript). Performance of the algorithms in the low-resolution spatial data across 75 samples of MOp with respect to Brier score and Jensen-Shannon divergence (lower better for both)

Figure R3 (Figure 3c in the manuscript). Distribution of performance of models on each individual spot in the low-resolution spatial data of developing human heart (top) and mouse SSp (bottom).

Figure R4 (Figure 4b in the manuscript). Boxplots of accuracy scores of the five methods applied to the 12 DLPFC slices across three experiment settings.

Comment #3: Other cell type deconvolution methods are also recommended for comparison:

- CytoSPACE (Vahid et al., Nature Biotechnology, 2023)
- TACCO (Mages et al., Nature Biotechnology, 2023)

Thank you for your suggestions. We have now referenced both TACCO and CytoSPACE in related work (now in Section 1). We have also included results of TACCO on our benchmark instances. The reviewer may note that CytoSPACE is not included in all plots as it was not possible to deploy and run CytoSPACE on some of our benchmark instances. Despite the significant amount of time and various computational resources that we dedicated to running this method, we faced several issues including: incompatibility issues with CytoSPACE’s linear assignment solvers, running out of memory (even with over 100GB allotted memory), issues with processing the input files (even though all other methods were able to successfully read/process the data), and the communication issues between CytoSPACE’s Python modules and its R modules (Seurat). Furthermore, given that both TACCO and CytoSPACE are of the “transportation” nature, we believe that this family of models is well represented in our benchmarks.

Please see Figures R1-R4 above for the quantitative comparison of DOT to these new cell type deconvolution methods. Please also see Figure R5 below for the performance of CytoSPACE and the number of instances solved by this method on our high-resolution benchmark instances.

Figure R5 (Figure A1 in Appendix C). First row shows accuracy values representing

the median of 64 values. Second row illustrates the number of instances (out of 64) solved by each model. All methods except CytoSpace and CARD solved all 64 instances across all experiments

Comment #4: The authors should benchmark DOT's computational requirements (including runtime and memory requirements) with a range of dataset sizes, rather than simply stating the results based on only four sets of data.

Thank you for your helpful suggestions. We have now performed extensive computational experiments to benchmark DOT's computational requirements with respect to runtime and memory requirements for several benchmark instances with different number of spots and different number of genes. We include supplementary figures (Figure A3, A4 and A5, also presented respectively in Figures R6, R7, and R8 below) that show detailed runtime information as a function of the number of spots and genes.

Figure R6 (Figure A3 in Appendix C). Total computation time of DOT for instances with different numbers of cells (spots) and genes in SRT.

Figure R7 (Figure A4 in Appendix C). Computation time of DOT as a function of number of spots (n) and number of genes (g).

Figure R8 (Figure A5 in Appendix C). Peak memory usage of DOT for instances of different sizes.

Comment #5: Most of the datasets used for benchmarking are generated by imaging-based techniques (MERFISH, osmFISH, and ISS) and contain only tens to hundreds of genes. I'm curious how DOT performs on datasets with more genes. To achieve this, the authors could utilize simulated SRT data constructed from scRNA-seq data (as implemented in the Cell2location) or constructed from Slide-seq data (as implemented in the CytoSPACE).

Thank you for your helpful comments. We would like to clarify that our simulated datasets for low-resolution mouse MOp (Figure 3b in Section 2.2) are already constructed from scRNA-seq data (as described in Section 4.3.3 in the manuscript). We use the MERFISH MOp data to inform the anticipated location of different cell types and assign cells from the scRNA-seq data to these locations based on the common cell type annotations. We then aggregate these cells to produce Visium-like spots. From this perspective, our approach is in the spirit of the procedure described in CytoSPACE, and, unlike the data simulation procedure implemented in cell2location, we don't use simulated data but rather information coming from real tissue samples.

We further elaborated on our simulation procedure to clarify this point in Section 4.3.3 as follows:

"We produced low resolution transcriptome-wide SRT samples based on the MERFISH MOp and scRNA-seq MOp (see Appendix D.1) as follows. Instead of placing cells from scRNA-seq MOp in random spatial locations, we used the MERFISH MOp data to guide the anticipated location of different cell types and assign cells from the scRNA-seq data to spatial locations based on the common "subclass" annotations between these two datasets. More specifically, for each of the 64 MERFISH MOp samples, we replaced each cell in the MERFISH MOp data with a randomly selected cell in the scRNA-seq MOp data of the same subclass. We then lowered the resolution of spatial data by splitting each sample into regular grids of length 100 μ m. Finally, we aggregated the expression profiles of cells within each tile to produce transcriptome-wide spots."

Comment #6: The authors should benchmark DOT's robustness on datasets with a range of spot resolutions (for example, mean of 5, 10, 20, ... cells per spot).

Thank you for your helpful comment. We would like to start by clarifying that DOT uses an "upper bound" on the size of spots in its optimization framework. This means that setting the upper bound to 20 does not imply that the average number of cells per spot is 20, rather DOT optimally determines the number of cells in each spot as part of its optimization process within this bound. To verify robustness of DOT to different upper bounds on the size of spots, we now show results of DOT under different spatial resolutions and different choices for DOT's upper bound parameter on the number of cells per spot in Figure A2 in Appendix C (also given in Figure R9 below).

For these experiments, we use the first six slices of the MERFISH MOp data in a manner similar to our experiments in Section 2.3, where we randomly assigned each cell from the MERFISH data to a cell of the same 'subclass' from the scRNA-seq data. We then aggregated the assigned cells at five levels of grid length (100 μ m, 125 μ m, 150 μ m, 175 μ m, and 200 μ m) to form multicell spots. The grid lengths correspond to approximately 10, 15, 20, 30, and 40 cells per spot on average, respectively (Figure R9a). We then assessed sensitivity of performance of DOT on decomposing the multicell spots into cell types as a function of spatial resolution (i.e., grid length and average number of cells per spot) and the upper bound parameter n , with $n \in \{10, 15, 20, 30, 40, 60, 100\}$. We observed that performance generally improves as resolution decreases (grid length increases), but, for a fixed spatial resolution, DOT exhibits a consistently high performance under different choices of parameter n (Figure R9b). We also verified that the number of cells per spot as estimated by DOT exhibits a strong correlation with the ground truth (i.e., number of cells within each grid) for different choices of n at different resolutions even when n is smaller than the expected number of cells per spot (Figure R9c).

Figure R9 (Figure A2 in Appendix C). Robustness of DOT under different spatial resolutions and different choices for DOT's upper bound parameter on the number of cells per spot (i.e., n).

Comment #7: I only saw the application of DOT on gene imputation. I suggest the authors provide some cell type deconvolution applications of DOT on real low-resolution SRT data (such as ST, 10x Visium, or Slide-seq).

Thank you for your suggestion. We have now added three more applications based on real Visium and ST data: spatial transcriptomics measurements from the

human dorsolateral prefrontal cortex, and human breast cancer (from two different studies). The results of our experiments on these applications are now included in Sections 2.4 and 2.5, respectively (Figures 4, 5, and 6 in the manuscript, also presented here in Figure R4 above and Figures R10 and R11 below).

Figure R10 (Figure 5 in the manuscript). Location and absolute abundance of cell types in three TNBC Visium samples. For each sample, we visualize the manual pathologist annotations (top left), number of cells per spot as estimated by DOT (bottom left), and enrichment of eight major cell types at their spatial locations as estimated by DOT.

Figure R11 (Figure 6 in the manuscript). Location and absolute abundance of cell types in two HER2+ breast tumors measured using Spatial Transcriptomics technology. For each sample, we visualize the morphological regions manually annotated by the pathologist into six categories: adipose tissue, breast glands, in situ cancer, connective tissue, immune infiltrate, and invasive cancer (top left). We also illustrate the number of cells per spot as estimated by DOT (bottom left), and enrichment of eight major cell types at their spatial locations as estimated by DOT (right).

- We have now included our experiments on the real Visium data from the human dorsolateral prefrontal cortex in Section 2.4 and have elaborated our experimental setup and results as follows:

“To evaluate performance on real low resolution SRT datasets and to demonstrate the ability of DOT in transferring spatial features beyond cell types/states, we next studied transferring layer annotations in the LIBD human dorsolateral prefrontal cortex (DLPFC) dataset [32]. This dataset contains spatial gene expression profiles of 12 DLPFC samples measured with 10X Visium, with the spots manually annotated with the six layers of the human DLPFC (L1 to L6) and white matter (WM). The samples correspond to two pairs of directly adjacent serial tissue sections from three independent neurotypical adult donors (i.e., four tissue sections per donor). In this experiment, we use DOT to transfer the layer annotations (L1 to L6, and WM) from one or a combination of reference Visium samples to a target Visium sample. Here, we use the reference Visium samples to characterize the expression profiles of the layers (i.e., without considering the spatial information in the reference samples, akin to our experiments in Section

2.2). Given that for each spot we know its true layer annotation (L1 to L6, and WM), transferring the layer annotations from other Visium samples to a target Visium sample allows us to truly quantify the accuracy of DOT (and other models) in determining the layer annotation of the spots in the target sample. Moreover, based on the reference Visium samples coming from the same or different donors, we can use this dataset to assess the accuracy of the models when the reference data is matched or unmatched.

To this end, we designed a total of 36 experiments categorized into three sets, where for each target Visium sample, we created three types of reference data for transferring the layer annotations to this sample. In the first set (denoted "adjacent" in Figure 4), for each of the 12 Visium samples, we determine the layer composition of a particular sample using its adjacent replicated Visium sample as the reference. In the second experiment (denoted "same brain" in Figure 4), we use the three Visium samples that belong to the same donor as the reference. Finally, in the third experiment (denoted 12 "aggregated" in Figure 4), we use all the 11 remaining Visium samples combined as the reference.

We compared the performance of DOT against four top performing methods from previous experiments (i.e., C2L, CARD, RCTD and TACCO) in Figure 4. For each experiment, we report the overall accuracy of the methods in terms of the percentage of the spots whose layers are correctly determined by each method. As illustrated in the boxplots and measured by the paired Wilcoxon signed-rank tests, DOT outperforms the benchmark methods with a statistically significant margin across all three experiments. In addition, while divergence from matched references (i.e., "adjacent" and "same brain") to an unmatched reference (i.e., "aggregated") affects the performance of all methods, DOT retains its performance above the baseline (i.e., 50% accuracy) in all 36 instances, with a median accuracy above 73% for matched reference and a median accuracy of 64% even for unmatched reference. All the while, the median accuracy for TACCO drops from 69% to 57%, and it drops to below 50% for C2L, CARD, and RCTD."

- We have also included our experiments on the real Visium/ST data from the human breast cancer in Section 2.5 and have elaborated our experimental setup and results as follows:

"For our second real low resolution SRT datasets, we analyzed a total of five human breast cancer samples coming from two independent studies. The first three samples coming from [33] are of the triple negative breast cancer (TNBC) tumor subtype and contain spatial gene expression profiles measured with 10X

Visium technology. The other two samples coming from [34] are of HER2+ tumor subtype and contain spatial gene expression profiles measured using Spatial Transcriptomics (ST) technology. Both datasets contain high level pathologist annotations (such as invasive cancer, cancer in situ, lymphocytes, immune infiltrate, and their mixtures) based on the H&E images. For both datasets, we used the scRNA-seq data coming from [33] as a reference to infer the cell type composition of the spots in each sample based on the matched tumor subtypes, and used the pathologist annotations to validate if the cell types are enriched in the anticipated locations.

We demonstrate localization of eight major cell types in the three TNBC Visium samples and the two HER2+ ST samples in Figures 5 and 6, respectively. In addition, we demonstrate the size of each spot (i.e., number of cells per spot) as estimated by DOT for each sample. Given that Visium offers a higher resolution (1-10 cells per sample on average) compared to ST platform (up to 200 cells per sample) [35], we set the upper bound parameter on the number of cells per spot (i.e., n ; see Section 2.1) for the Visium and ST samples to 20 and 200, respectively, and observe that DOT determines the number of cells per spot consistently with the expected density of different regions. Moreover, denser regions are enriched in smaller cells (such as lymphocytes/immune cells), while larger cells (such as adipose and stromal cells) appear more frequently in regions with low density. Of note, DOT correctly localized cancerous epithelial cells in accordance with the respective pathologist annotations (e.g., invasive cancer and cancer in situ) and normal epithelial cells in accordance with normal cells (see sample CID44971 in Figure 5). Moreover, DOT localizes T/B cells in accordance with lymphocytes and immune cells and their combinations with other cells (e.g., invasive cancer), which, as expected, are enriched in the vicinity of tumor cells [36].”

Minor:

The description of “Sub population” is confusing. I suggest the authors revise it to “Cell population”.

Thank you for your suggestion. We agree and have now used “cell population” instead of “sub population” consistently across the manuscript.

Reviewer #2 (Remarks to the Author):

The authors proposed a novel tool, DOT, for integrating single-cell and spatial transcriptome in this manuscript. DOT attempts to match the gene expressions between single cell and spatial transcriptome, cell subpopulation abundance and capture spatial relations by a multi-objective optimization model. The idea of DOT is interesting but the manuscript is not well written and the evaluation of the performance is also not well designed and the results are not convincing for me.

Comment #1: In “DOT locates cell types in high-resolution spatial data” section, DOT is compared with RCTD, Seurat, Tangram and SingleR. RCTD is designed for cell type decomposition of spatial transcriptomics such as slide-seq and Visium, not for the image-based method such as MERFISH which contains only hundreds of genes. RCTD should not be included in this scenario. In addition, some popular integration and transferring methods such as harmony, TACCO and Spatial-ID should be included.

Thank you for your suggestion. We had included RCTD since even though it is developed for low-resolution, it showed good performance in high resolution as well. For consistency, we have now removed RCTD from these experiments and added TACCO, Harmony, and NovoSpaRc instead (please see Figure 2 in Section 2.2 and Figure A1 in Appendix C, also given in Figures R1 and R5 above). We did also look into Spatial-ID, but could not find a reproducible package or guidelines for setting up custom experiments beyond their pretrained models, so that we were unfortunately not able to include it in our benchmark.

Comment #2: In “DOT determines cell type abundances in low-resolution spatial data” section, the authors evaluated the performance of decomposition of spatial transcriptomics. Decomposition is only used for low resolution low-resolution SRT such as Visium which includes the whole transcriptome. But the datasets used in the evaluation are produced from MERFISH and osmFISH which include only 254 genes and 33 genes, respectively. So these two datasets are totally NOT suitable for the decomposition. The results are meaningless. Some other deconvolution methods such as CARD should be included here as well.

Thank you for your helpful comments. We would like to clarify that our simulated datasets for low-resolution mouse MOp (Figure 3b in Section 2.2) are already constructed from scRNA-seq data (as described in Section 4.3.3 in the old manuscript). We use the MERFISH MOp data to inform the anticipated location of

different cell types and assign cells from the scRNA-seq data to locations based on the common cell type annotations. We then aggregate these cells to produce Visium-like spots. We have now clarified this point in the revised manuscript.

We included five new models in our benchmark: TACCO, CARD, NovoSpaRc, CytoSPACE and Harmony (please see response to Comment 3 of reviewer 1).

In addition, we have also included three new applications based on real Visium and ST data: spatial transcriptomics measurements from the human dorsolateral prefrontal cortex, and human breast cancer (from two different studies). The results of our experiments on these applications are now included in Sections 2.4 and 2.5, respectively (Figures 4, 5, and 6 in the manuscript, also presented here in Figures R4, R10 and R11 above). Please see our response to Comment #7 of reviewer 1 for more details on these experiments.

Comment #3: DOT is a multiple objective optimization model which have several weights parameters for objectives. How do these user-defined penalty weights effect the performance?

While DOT offers the users with the flexibility to set these parameters, by default DOT derives these weights directly from the data in such a way that all metrics contribute roughly equally to the objective. This is to make the method accessible to the users from all domains. We incorporated your comment by performing extensive parameter tuning analysis (with over 100,000 runs) and have included the results in the revised manuscript in Figure A6 in Appendix C (also presented in Figure R12 below).

Thanks to your helpful comment, we have now found that slight changes in our default parameters may prove useful in a more general setting and beyond our initial result reported in our first submission. We have now reflected these parameter choices in our package implementation as well.

Figure R12 (Figure A6 in Appendix C). Top performing parameter settings for the simulated low resolution MOp and high resolution MOp with different numbers of genes.

Comment #4: The Methods section is too long. Section 4.1 “related work” should be moved to Introduction and some mathematical details should be moved to supplementary.

Thank you for your suggestion. We have now removed “related work” by moving parts of what was previously in “related work” to the introductory section, and some parts to Appendix B, to shorten the Methods section, and clearly state the contributions of DOT and the limitations of the previous work it addresses.

Reviewer #1 (Remarks to the Author):

All my comments are addressed in the revision. I only have one minor suggestion:

1. I suggest the authors provide the benchmarking codes on GitHub for other readers to better reproduce the results presented in the manuscript.

Reviewer #1 (Remarks on code availability):

I believe the code is OK and is capable of generating the results presented in the manuscript. The README file has included enough instructions for installing and running the application.

Reviewer #2 (Remarks to the Author):

In the revised manuscript, the authors added RCTD and TACCO for comparison and three real low resolution datasets for evaluation. Although there has been improvement compared to the previous version, I still have some concerns about the validation.

1. In Section 2.2, the authors tried to "transfer cell types from single cell to high resolution SRT". This is not deconvolution or decomposition since MERFISH has sub cellular resolution. Deconvolution means inferring the cell type information of the spots which are mixed by the MULTIPLE CELLS with different cell types. Here the authors replace RCTD with CARD for comparison. I am glad that authors take my suggestion to remove RCTD here because RCTD is designed for deconvolution NOT for transferring cell type annotations. BUT CARD IS DESIGNED FOR DECONVOLUTION AS WELL. CARD should not be included in this section, too.

2. In Section 2.5, the authors analyzed two low resolution real datasets and showed the results from DOT. However, I can not see any advantages of DOT from these results. There is no any comparison with other methods or any new findings different from previous studies. Considering there are only three low resolution real datasets, I think the valuation of DOT is not comprehensive enough.

3. The author claims "DOT works for both absolute counts and normalized expressions" as one of their distinctive features. But why do we want this feature? Absolute counts can be normalized easily. The tool which works for either absolute counts or normalized expressions is enough.

Reviewer #2 (Remarks on code availability):

The code is fine and there are enough instructions for installing and running the application.

We would like to thank the reviewers for their constructive comments. Below, we provide their comments in *italic* typeface and our responses in normal typeface.

Reviewer #1 (Remarks to the Author):

All my comments are addressed in the revision. I only have one minor suggestion:

1. I suggest the authors provide the benchmarking codes on GitHub for other readers to better reproduce the results presented in the manuscript.

Thank you again for your helpful comments and suggestions throughout the revision process. We now made the code needed to reproduce the results freely available on GitHub (https://github.com/saezlab/dot_experiments), that we point at now from the manuscript.

Reviewer #2 (Remarks to the Author):

In the revised manuscript, the authors added RCTD and TACCO for comparison and three real low resolution datasets for evaluation. Although there has been improvement compared to the previous version, I still have some concerns about the validation.

Thank you again for your helpful comments and careful reading of the paper. Please see below our point to point responses to your comments.

Comment #1: *In Section 2.2, the authors tried to "transfer cell types from single cell to high resolution SRT". This is not deconvolution or decomposition since MERFISH has sub cellular resolution. Deconvolution means inferring the cell type information of the spots which are mixed by the MULTIPLE CELLS with different cell types. Here the authors replace RCTD with CARD for comparison. I am glad that authors take my suggestion to remove RCTD here because RCTD is designed for deconvolution NOT for transferring cell type annotations. BUT CARD IS DESIGNED FOR DECONVOLUTION AS WELL. CARD should not be included in this section, too.*

Thank you for the suggestion. We appreciate this point and we have now removed CARD from Section 2.2 and Figure 2 and don't include any other deconvolution approach.

Just for clarification, the reason we had included CARD is because transferring labels or mapping other single variables whether categorical or continuous, from the perspective of mathematical optimization and mapping, is a specialization of the same task. In particular, deconvolution is a many-to-many mapping while label transfer is a many-to-one mapping. Both domains of the input and output spaces retain their characteristics in both tasks. That being said, we agree that it can't be expected that approaches designed specifically to address the many-to-many task will be applicable to the many-to-one task, and as stated above we have now removed CARD.

Comment #2: *In Section 2.5, the authors analyzed two low resolution real datasets and showed the results from DOT. However, I can not see any advantages of DOT from these results. There is no any comparison with other methods or any new findings different from previous studies. Considering there are only three low resolution real datasets, I think the valuation of DOT is not comprehensive enough.*

Regarding your comment about the comparison of DOT with other methods in Section 2.5, we have now updated Figure 5 (see below), where we compare the deconvolution results from DOT with those from four top performing methods (CARD, Cell2location, RCTD, and TACCO) and highlight robustness of DOT in reliably locating major cell types according to pathological annotations.

We would like to also clarify and emphasize the comprehensiveness of our evaluations in all analyses and benchmarks, that include the three data sets mentioned by the reviewer, as well as simulation data. Our results include comparison to 10 related methods, and are based on a total of 100,000 runs of our approach on this data. Let us first point out that there are no ground truth cell-type distributions available in low-resolution spatial data due to the limitation of low-resolution technologies. We addressed this limitation by performing objective evaluation and comparative analysis based on (i) simulated low-resolution spatial data (Section 2.3), (ii) transferring spatial features beyond cell types (Section 2.4), and (iii) pathological annotation as a proxy for expected enrichment regions of cell types (Section 2.5). In particular:

- ***Simulated low-resolution spatial data (Section 2.3).*** To produce quantifiable objective ground truth for low-resolution, we used the cell-type information from MERFISH, osmFISH, and ISS from different organs and datasets. On the MERFISH dataset, we compared the results of DOT against 10 different deconvolution approaches in 64 experiments. To ensure the relevance of the findings and the sensitivity of the performance of DOT we also performed over 100,000 runs, varying resolution and cell density (Figure R9a). On the latter two datasets, we showed the cell-type distributions resulting from DOT and compared them with those from 10 other approaches on each individual bin.
- ***Transferring spatial features beyond cell types (Section 2.4).*** On the human dorsolateral prefrontal cortex dataset, using a ground truth from manual layer annotations of the samples we defined a novel task of transferring spatial features beyond cell types. To the best of our knowledge, this is the first illustration of the transfer of high-level features between slides from the same condition from transcriptomics data and it has implications for adding confidently additional layers of annotation to samples measured with spatial omics facilitating downstream analyses. To ensure quality transfer of annotations with DOT we designed 36 scenarios of transferring annotations from consecutive slides, from non-consecutive slides coming from the same donor and from aggregated resources of slides from the same organ. We compared the performance of DOT on all tasks to the four top-performing approaches from previous tasks. DOT outperforms all approaches on all tasks with a statistically significant margin.
- ***Pathological annotation as a proxy for expected enrichment regions of cell types (Section 2.5).*** As a third approach, we used pathological annotation of the regions containing

information about the observed cell type composition per region on the two independent breast cancer datasets (5 experiments in total) as a proxy for ground truth. On these two datasets, we showed that DOT correctly estimates the cell-type density and assigns cell types matching the observations from the ground truth pathological annotation and further provides richer cell-type distribution, matching regional expectations. To further highlight the robustness of DOT, we now compare the cell type deconvolution results for DOT with four other top performing methods (CARD, Cell2location, RCTD, and TACCO). For DOT and Cell2location, we present both absolute and relative abundances; for the other methods we present relative abundances. These results highlight the robust performance of DOT in locating major cell types according to pathological annotations. In particular, DOT correctly locates cancerous and normal epithelial cells as well as T cells, whereas other methods tend to over- or under-estimate cancer cells (such as Cell2location and CARD), are not robust in detecting T cells, and predominantly underestimate normal epithelial cells. The new Figure 5 (also presented below) shows these results. Note that in light of the new results we changed the order of presenting the results in Section 2.5. We now start with the results from the HER2+ study and continue with the results from the TNBC study (now Figure 6).

To clarify this, we also discuss these experiments in the Discussion section of the revised manuscript.

Comment #3: *The author claims "DOT works for both absolute counts and normalized expressions" as one of their distinctive features. But why do we want this feature? Absolute counts can be normalized easily. The tool which works for either absolute counts or normalized expressions is enough.*

Thank you for your comment and we apologize for any confusion; when we wrote "with both absolute counts and normalized expressions" we meant "with both discrete counts and continuous expressions", which is how we have updated our manuscript. We would like to clarify that our goal here is to draw a contrast between discrete and continuous measurements. This is relevant since statistical methods, such as RCTD and cell2location, base their statistical hypotheses on the discrete nature of mRNA counts in both scRNA-seq and spatial data, and that is why they are not applicable to continuous measurements. You are right in that all discrete measurements can be normalized; however, the discrete counts might not always be available. Here, by normalized expressions, we mean continuous expressions (as opposed to discrete expressions), which can result from standard normalization techniques applied to discrete count data, technology-specific normalization methods (such as normalization by cell volume in hybridization-based technologies), or probabilistic methods that yield gene density rather than segmentation-based count data. We have now clarified this point in the revised manuscript by changing "with both absolute counts and normalized expressions" to "with both discrete counts and continuous expressions".

Reviewer #2 (Remarks to the Author):

Thank the authors for addressing all my concerns. Only one minor:

1. At the end of section 2.5, "In comparison, other methods tend to over- or under-estimate cancer cells, are not robust in detecting T cells, and predominantly underestimate normal epithelial cells (Figure 5B)." Figure 5B should be Figure 5C.

Reviewer #2 (Remarks on code availability):

The code is good.